

# Assessment of Sub-Shelf Melting Parameterisations Using the Ocean-Ice Sheet Coupled Model NEMO(v3.6)-Elmer/Ice(v8.3)

Lionel Favier[1], Nicolas C. Jourdain[1], Adrian Jenkins[2], Nacho Merino[1], Gaël Durand[1], Olivier Gagliardini[1], Fabien Gillet-Chaulet[1], and Pierre Mathiot[3]

[1]Univ. Grenoble Alpes, CNRS, IRD, IGE, 38000 Grenoble, France
[2]British Antarctic Survey, Cambridge, CB3 0ET, UK
[3]Met Office, Exeter, UK

**Correspondence:** Lionel Favier (lionel.favier@univ-grenoble-alpes.fr)

**Abstract.**

Oceanic melting beneath ice shelves is the main driver of the current mass loss of the Antarctic ice sheet, and is mostly parameterised in stand-alone ice-sheet modelling. Parameterisations are crude representations of reality, and their response to ocean warming has not been assessed in regard to 3D ocean-ice sheet coupled models. Here, we assess various melting

parameterisations ranging from simple scalings with far-field thermal driving to emulators of box and plume models, using a new coupling framework combining the ocean model NEMO and the ice-sheet model Elmer/Ice. We define six idealised one-century scenarios for the far-field ocean ranging from cold to warm, and representative of potential futures for typical Antarctic ice shelves. The scenarios are used to constrain an idealised geometry of the Pine Island glacier representative of a relatively small cavity. Melt rates and sea-level contributions obtained with the parameterised stand-alone ice-sheet model are

compared to the coupled model results. The plume parameterisation underestimates the contribution to sea level when forced by the warm(ing) scenarios. The box parameterisation compares fairly well to the coupled results in general and gives the best results using five boxes. For simple scalings, the comparison to the coupled framework shows that a quadratic dependency to thermal forcing is required, as opposed to linear. In addition, the quadratic dependency is improved when melting depends on both local and nonlocal, i.e. averaged over the ice shelf, thermal forcing. The results of both the box and the two

quadratic parameterisations fall within or close to the coupled model uncertainty. Considering more robust sub-shelt melting parameterisations is key to decrease uncertainties on the Antarctic contribution to sea level rise. Comparing parameterisations to ocean/ice-sheet coupled simulations under various scenarios helps to assess them.

## 1 Introduction

The majority of grounded ice in Antarctica is drained through its floating extensions advancing in the Southern Ocean. The

increase of ice-shelf thinning since the 1990s has been mostly driven by ice mass loss from the western part of the ice sheet (Paolo et al., 2015; Shepherd, 2018). In the Amundsen and Bellingshausen seas, ice-shelf thinning is due to incursions of Circumpolar Deep Water (CDW) beneath the ice-shelf base all the way to the line-boundary between the grounded and floating part of the ice sheet, i.e. the grounding line. These incursions episodically increase the ocean-ice heat flux and drive sub-shelf





melting and ice-shelf thinning (Jacobs et al. (2011); Dutrieux et al. (2014); Jenkins et al. (2018) for West Antarctica and Gwyther et al. (2018) for East Antarctica). The thinning of floating ice decreases the backforce restraining the upstream ice, leading to ice-sheet acceleration (Mouginot et al., 2014), depressing grounded ice surface (Konrad et al., 2017), retreating grounding lines (Rignot et al., 2014; Konrad et al., 2018) and eventually increased sea level rise.

West Antarctic grounding lines often rest on retrograde bed upsloping towards the ocean (Fretwell et al., 2013). This makes the glaciers vulnerable to the Marine Ice Sheet Instability (MISI), which states that an ice sheet starting to retreat over a retrograde bed slope keeps retreating until the slope becomes prograde (Mercer, 1978; Thomas and Bentley, 1978; Weertman, 1974; Schoof, 2007; Durand et al., 2009). Confined ice shelves resist to horizontal shearing and potentially stabilise an ice sheet undergoing a MISI (Gudmundsson et al., 2012; Gudmundsson, 2013; Haseloff and Sergienko, 2018). Ice-sheet modelling
results suggest that the Pine Island and the Thwaites glaciers may have started an unstable retreat (Favier et al., 2014; Joughin et al., 2014), but the tipping point beyond which a MISI occurs is not clearly identified (Pattyn et al., 2018).

Ocean warming is currently the main driver of the West Antarctic ice sheet retreat, and can potentially trigger further MISI (Favier et al., 2014; Joughin et al., 2014). Using realistic ice-shelf basal melt rates in ice-sheet simulations is therefore crucial. The most comprehensive way to do so consists of using an ocean model that solves the 3D Navier-Stokes equations in ice-
shelf cavities and represents ocean-ice heat exchanges (Losch, 2008). The existence of strong feedbacks between the cavity geometry, melt rates, and the ocean circulation (De Rydt et al., 2014; Donat-Magnin et al., 2017) has motivated the development of coupled ocean-ice sheet models presenting a moving ocean-ice boundary. To date, this kind of coupled models has been used in idealised configurations (e.g. De Rydt and Gudmundsson, 2016; Asay-Davis et al., 2016; Jordan et al., 2018; Goldberg et al., 2018) or with more realistic configurations representing a single ice shelf (Thoma et al., 2015; Seroussi et al., 2017). However,
the required numerical developments and the relatively high computational cost of the ocean component strongly limit the use of ocean-ice coupled models for long term simulations of the Antarctic ice sheet.

A much simpler approach to account for oceanic forcing in stand-alone ice-sheet models is to prescribe melting by plugging off-line ocean model outputs (e.g. Seroussi et al., 2014). The melt rates cannot evolve with cavity geometry changes. Mengel and Levermann (2014) improved the method by correcting the dependency of the freezing point to a changing ice-draft, but
it is still unable to account for the dependency to far-field temperature and salinity stratification, and for circulation changes driven by the evolution of the cavity geometry (Donat-Magnin et al., 2017). This approach also requires the choice of empirical ad-hoc melt rates underneath newly floating ice wherever the grounding line is retreating during the prognostic simulations. To circumvent this issue, Cornford et al. (2015) and Nias et al. (2016) consider the ice mass flux near and away from the grounding line to build a sound initial melting pattern that depends on the distance to the grounding line and adapts to its further migration.
By construction, the melt rates are much larger at the grounding line and decrease exponentially away from it. Spatially and temporally varying melt rates (anomalies) taken from ocean models are added to these initial melt rates to predict future sea level contribution. This latter approach is also empirical and does not account for potential change in oceanic circulation (e.g. due to feedbacks with ice dynamical changes).

The melt rates can also be parameterised using two main approaches being either an explicit function of depth or a function
depending on far-field ocean temperature and salinity. In the first approach (followed by, e.g. Favier et al., 2014; Joughin et al.,





2014, with more examples given in Asay-Davis et al. (2017)), they are computed by a piecewise linear function of depth and an initial calibration is done to match current observations on average (e.g. using datasets from Rignot et al., 2013b; Depoorter et al., 2013). The oversimplicity of the depth-dependence not only makes the initial pattern far from the observed pattern, but also leads to a significant overestimation of the grounding-line retreat compared to ocean-ice sheet coupled models (Seroussi
et al., 2017; Jordan et al., 2018; De Rydt and Gudmundsson, 2016).

The second approach parameterises the melt rates as a function of ocean temperature and salinity profiles. The simplest parameterisations are mere functions of the difference between the temperature and the melting/freezing point at the ice-ocean boundary, the thermal forcing, using a linear (e.g. Beckmann and Goosse, 2003; Favier et al., 2016) or a quadratic dependency (e.g. DeConto and Pollard, 2016). More complexity is accounted for in the box model proposed by Reese et al. (2018a) and
based on the 1D ocean-box model from Olbers and Hellmer (2010), and also in the 2D emulation of a 1D plume model (Jenkins, 1991) proposed by Lazeroms et al. (2018).

Assessing these last parameterisations in regard to melt rates computed by a stand-alone ocean model would enable to investigate the patterns differences in a static cavity geometry. However, the melt-rates pattern has also an effect on the ice-sheet response. The study of Gagliardini et al. (2010) highlights configurations where less melting leads to a grounding line
relatively further upstream, or where the same average melting leads to two different ice-sheet responses and grounding-line positions. An ice-sheet model is therefore needed to carry out a meaningful comparison between parameterized and simulated melt rates.

In this paper, we assess several flavours of the aforementioned ocean temperature and salinity dependent parameterisations in regard to ocean-ice sheet coupled simulations. We include the uncertainties arising from the ocean model by considering an
ensemble of four ocean-ice coupled configurations. Following an initial calibration that allows further comparisons between parameterised and coupled simulations, we use six one-century far-field ocean temperature and salinity scenarios, which we apply to drive the melting parameterisations in stand-alone ice sheet simulations and force the members of the ocean ensemble in ocean-ice sheet coupled simulations. Overall, the MISOMIP (Asay-Davis et al., 2016) framework is used to perform 138 one-century simulations (19 sub-shelf melt parameterisations + 4 coupled members × 6 scenarios).

The paper is organised as follows. The next section describes the models: the ice-sheet model Elmer/Ice, the ocean model NEMO and the framework for coupling those two models. The section also describes the sub-shelf melt-rates parameterisations and the members of the ocean-ice ensemble. The third section describes the experiments, including the reference setup of the ocean-ice sheet system, the initial calibration of the parameterised and coupled simulations and the set of far-field ocean temperature and salinity scenarios. Then, we detail the results in regard to sea-level contribution and sub-shelf melting evolution,
and discuss the use of sub-shelf melt parameterisations in stand-alone ice sheet modelling at a regional or a global scale.



## 2   Models

### 2.1   The ice-sheet model, Elmer/Ice

We perform the ice-sheet simulations with the finite-element ice-sheet model Elmer/Ice (Gagliardini et al., 2013). The ice rheology is non linear and controlled by the Glen's flow law, enabling to link the deviatoric stress tensor and the strain rate

tensor from which ice velocities are retrieved. The used version of the ice-sheet model solves the SSA* solution, a variant of the L1L2 solution of Schoof and Hindmarsh (2010) solving the shallow shelf approximation of the Stokes equations and accounting for vertical shearing in the effective strain rate. The SSA* approximation was recently implemented in Elmer/Ice following the work of Cornford et al. (2015).

To calculate the basal friction, the grounding line position is calculated from hydrostatic equilibrium and can thus be located

anywhere within an element. We use a sub-element parameterisation to affect basal friction to the part of the element that is grounded by increasing its number of integration points (Seroussi and Morlighem, 2018, equivalent to the SEP3 method in). The basal friction is computed by a Schoof-like friction law based on the theoretical work of Schoof (2007) applied to a linear ice rheology, and which was extended to a non-linear rheology by Gagliardini et al. (2007). The Schoof friction law (equation written in Appendix A) depends on the difference between ice overburden pressure and the water pressure, i.e.

the effective pressure, and exhibits two asymptotic behaviours. The law behaves as a non-linear power law away from the grounding line and as a Coulomb friction law near the grounding line, thus ensuring a smooth transition of stress state near and at the grounding line. The Schoof friction law was recently compared to various other types of friction laws commonly used in ice-sheet modelling, for an idealised framework (Brondex et al., 2017) and a real drainage basin (Brondex et al., 2018).

There is no sub-element parameterisation to calculate basal melting, which is applied to floating nodes only.

The mesh grid is unstructured and made of triangles, the size of which is about $500\ m$ in the vicinity of the grounding line and up to $4\ km$ away. The Elmer/Ice configuration is identical for parameterised and coupled simulations.

### 2.2   Ocean melting from a 3D ocean-ice sheet coupled model

The melt rates beneath the ice shelf are either parameterised or computed through the coupling of NEMO and Elmer/Ice. Here we describe the ocean model and the ocean-ice sheet coupling framework.

### 2.2.1   The ocean model, NEMO

We make use of the 3D primitive-equation ocean model NEMO-3.6 (Madec and NEMO-team, 2016, Nucleus for European Modelling of the Ocean). NEMO solves the prognostic equations for the ocean temperature, salinity and velocities, and includes ice-shelf cavities (Mathiot et al., 2017). The sub-shelf melting is parameterised through the so-called "three equations" representing (1) the heat balance at the ice-ocean interface accounting for phase change, turbulent exchange in water, and dif-

fusion in the ice; (2) the salt balance accounting for freezing/melting and turbulent exchange; and (3) the pressure and salinity dependence of the potential temperature at which seawater freezes (Hellmer and Olbers, 1989; Holland and Jenkins, 1999;





Losch, 2008; Jenkins et al., 2010). In this parameterisation, we assume a constant top-boundary-layer (TBL) thickness along the ice-shelf draft, and we use a velocity-dependent formulation in which the heat exchange velocity is defined as:

$$\gamma_T = \Gamma_T \sqrt{C_d(u_{\text{TBL}}^2 + u_{\text{tide}}^2)} \tag{1}$$

where $u_{\text{TBL}}$ the TBL-averaged velocity resolved by NEMO, $\Gamma_T$ is the non-dimensional heat exchange coefficient, $C_d$ the non-dimensional drag coefficient and $u_{\text{tide}}$ a uniform background velocity representing the main effect of tides on ice-shelf melting (Jourdain et al., 2018). The values of $\Gamma_T$, $C_d$ and $u_{\text{tide}}$ are given in Tab. 1.

The ocean configuration used in this study is very similar to the ISOMIP+ configuration described by Asay-Davis et al. (2016): we use a linearised equation of state and the only lateral boundary condition is a temperature and salinity restoring along the vertical boundary representing offshore conditions; neither sea ice nor atmospheric forcing nor tides are represented. The only differences with the general MISOMIP protocol is that we use different temperature and salinity restoring and initial conditions. We use a variety of resolutions and parameters for NEMO to build an ensemble of NEMO-Elmer/Ice coupled simulations as described in section 2.2.3.

### 2.2.2 The ocean-ice sheet coupled model framework

We couple NEMO and Elmer/Ice, meaning that Elmer/Ice sees sub-shelf melt rates calculated by NEMO, while NEMO sees the ice-shelf geometry resulting from the ice dynamics resolved by Elmer/Ice. A given coupling period (typically of few months) is first covered by the ocean model with a cavity geometry from the end of the previous coupling period; then, the period is covered by the ice-sheet model forced by the oceanic melt rates averaged over this coupling period (Fig. 1).

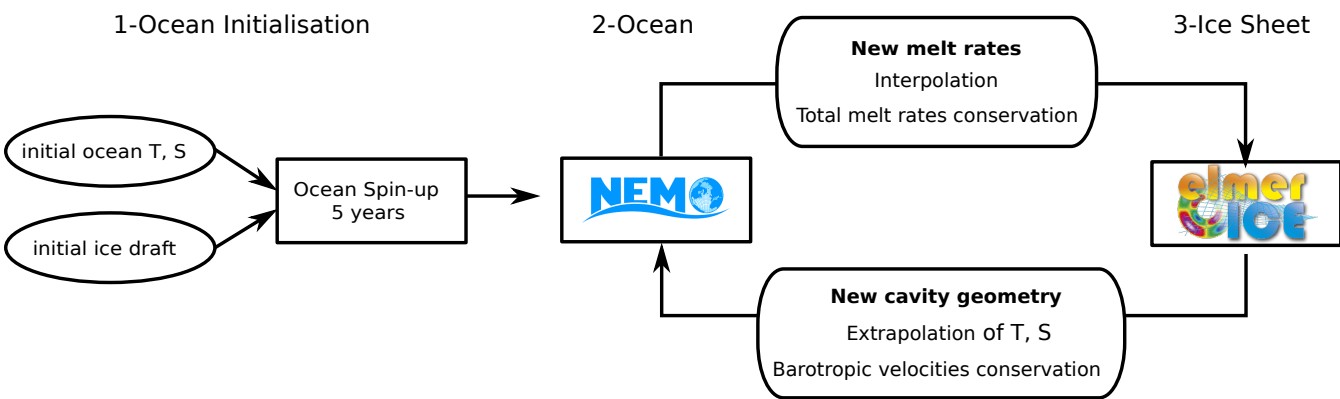

**Figure 1.** NEMO-Elmer/Ice coupling framework. T and S stand for temperature and salinity.

As the respective grids of the two models differ, some interpolation is required for each exchange. Following each NEMO run, Elmer/Ice restarts from its previous time step (ice geometry and velocities). The melt rates provided by NEMO are bi-linearly interpolated onto Elmer/Ice's unstructured grid. A multiplicative correction factor computed over the entire ice shelf ensures that the same mass flux is seen by the two models (this factor is very close to one in our case). In case Elmer/Ice has a floating element but the water column is too thin to be captured by NEMO, the melt rate seen by Elmer/Ice is set to zero.

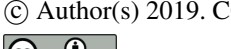



Every coupling period, NEMO restarts with temperature, salinity and velocities from its previous time step using the updated geometry from Elmer/Ice. If new ocean cells appear (previously masked ice cells), temperature and salinity are an average of the four closest wet cells (horizontally if possible, vertically extrapolated otherwise), and ocean velocities are set to zero. To avoid the generation of spurious barotropic waves as a result from sudden changes in water column thickness, we impose a

5 conservation of barotropic velocities across the step-change in the ice-shelf geometry. We also conserve the sea surface height (SSH) value for all the water columns, and if a new water column is created, SSH is an average of the four closest wet cells.

We use the same initial state for Elmer/Ice as in MISOMIP (Asay-Davis et al., 2016), i.e. a steady state obtained with zero melt, and NEMO is spun up for 5 years with this initial ice-shelf geometry before being coupled to Elmer/Ice. The respective time steps of Elmer/Ice and NEMO are 1 month and 200 s, and the coupling period ranges between 2 and 6 months, depending

10 on the configuration. Sensitivity studies undertaken following the MISOMIP protocol (Asay-Davis et al., 2016) indicate very little sensitivity to coupling periods between 1 month and 1 year, with less than 3% difference in sea-level contribution after 100 years (Figure A1).

### 2.2.3 The ensemble of ocean configurations within the coupled framework

While the NEMO ocean model is much more representative of the ocean physics than any sub-shelf melting parameterisation,

15 there are still processes like turbulence and convection that need to be parameterised. The model is also sensitive to both the horizontal and vertical resolutions. To account for the consequent ocean model uncertainty, we consider four NEMO configurations with the varying parameters listed in Tab. 1. For each coupled configuration, the $\Gamma_T$ parameter is adjusted following the exact ISOMIP+ calibration protocol after a 4 years of ocean spin up with a steady ice-shelf draft (Asay-Davis et al., 2016).

**Table 1.** Ocean parameters used for the four NEMO-Elmer/Ice coupled simulations.

| ID | Name | $\Delta x$ (km) | $\Delta z$ (m) | $T_{CPL}$ (month) | $\Gamma_T$ ($\times 10^{-2}$) | $u_{tide}$ (m.s$^{-1}$) | $K_{stab}$ $\nu_{stab}$ | $K_{unstab}$ $\nu_{unstab}$ |
|---|---|---|---|---|---|---|---|---|
| 1 | COM | 2.0 | 20.0 | 6 | 4.00 | 0.01 | uniform | 0.1 m$^2$.s$^{-1}$ |
| 2 | COM-tide | 2.0 | 20.0 | 6 | 3.15 | 0.05 | uniform | 0.1 m$^2$.s$^{-1}$ |
| 3 | TYP-1km | 1.0 | 20.0 | 2 | 4.00 | 0.01 | TKE param. | 10 m$^2$.s$^{-1}$ |
| 4 | TYP-10m | 2.0 | 10.0 | 3 | 9.60 | 0.01 | TKE param. | 10 m$^2$.s$^{-1}$ |

$\Delta x$ is the horizontal resolution, $T_{CPL}$ is the ocean-ice sheet coupling period and $\Delta z$ is the nominal vertical resolution. The actual resolution near the sea floor or ice shelf draft can be smaller due to the use of partial steps, but the TBL thickness is always equal to $\Delta z$ (i.e. TBL quantities are averaged over several levels in the case of partial steps). $\Gamma_T$ and $u_{tide}$ are defined in Eq. (1), and the salt exchange coefficient $\Gamma_S$ is taken as $\Gamma_T/35$. Also defined in Eq. (1), the drag coefficient $C_d = 2.5 \times 10^{-3}$. The stable vertical diffusivity and viscosity coefficients ($K_{stab}$ and $\nu_{stab}$ respectively) are either constant, at the same values as in Asay-Davis et al. (2016), or calculated through the TKE scheme with the same parameter values as in Treguier et al. (2014). Convection is parameterised through enhanced diffusivity and viscosity ($K_{unstab}$ and $\nu_{unstab}$ respectively) in case of static instability (0.1 m$^2$ s$^{-1}$ as Asay-Davis et al. (2016) and 10 m$^2$ s$^{-1}$ as Treguier et al. (2014)). The remaining parameters are exactly the same as in the common ISOMIP+ configuration described in Asay-Davis et al. (2016).



### 2.3 Ocean melting from ocean-dependent sub-shelf parameterisations

All the parameterisations are linked to ambient temperature and salinity vertical profiles in the far-field ocean. The stand-alone ice-sheet simulations start from the same initial state as for the ocean-ice sheet coupled simulations. The parameterisations respond instantaneously to changes in ambient temperatures and salinities, i.e. they do not account for ocean circulation time

scales (e.g. water residence time in ice-shelf cavities, Holland, 2017). None of the parameterisations account for the Coriolis effect or for bathymetric features (e.g. sills, channels). To avoid areas of very thin ice that would affect the stability of the ice-sheet model, melting is not permitted wherever the ice base is shallower than 10 m depth.

#### 2.3.1 Simple functions of thermal forcing

The following three parameterisations are based on an expression for the ice-ocean heat transfer that is analogous to the one

used in more complex ocean circulation models (Grosfeld et al., 1997). However, they make the simplifying assumption that the thermal forcing across the ice-ocean boundary layer can be determined directly from far-field ocean conditions. Thus, cooling of the water as it is advected from the far-field into the cavity and then mixed into the ice-ocean boundary layer is accounted for simply through the choice of an effective heat transfer coefficient.

**The linear, local dependency** to thermal forcing assumes a balance between vertical diffusive heat flux across the ocean cavity top boundary layer and latent heat due to melting-freezing. Its formulation is based on Beckmann and Goosse (2003) and written as:

$$M_{lin} = \gamma_T \frac{\rho_{sw} c_{po}}{\rho_i L_i} (T_o - T_f). \tag{2}$$

with $\gamma_T$ the heat exchange velocity (aimed at being calibrated, see Sec. 3.2), $\rho_{sw}$ and $\rho_i$ the respective densities of ocean

water and ice, $c_{po}$ the specific heat capacity of the ocean mixed layer and $L_i$ the latent heat of fusion of ice (Tab. 2). The melting-freezing point $T_f$ at the interface between the ocean and the ice-shelf basal surface is defined as:

$$T_f = \lambda_1 S_o + \lambda_2 + \lambda_3 z_b. \tag{3}$$

The practical salinity $S_o$ and the potential temperature $T_o$ are taken from the far-field ocean as detailed in this Sec. 2.3.1, $z_b$ is the ice base elevation, which is negative below sea level, and the coefficients $\lambda_1$, $\lambda_2$, $\lambda_3$ are respectively the liquidus slope,

intercept and pressure coefficient.

    The linear formulation with a constant exchange velocity assumes a circulation in the ice-shelf cavity that is independent from the ocean temperature. This assumption is neither supported by modelling (Holland et al., 2008; Donat-Magnin et al., 2017) nor by observational (Jenkins et al., 2018) studies that suggest a more vigorous circulation in response to a warmer ocean, subsequently increasing melt rates.

    **The quadratic, local dependency** to thermal forcing accounts for this positive feedback between the sub-shelf melting and the circulation in the cavity (Holland et al., 2008), using a heat exchange velocity linearly depending on local thermal forcing.



The formulation is written as:

$$M_{quad} = \gamma_T \left( \frac{\rho_{sw} c_{po}}{\rho_i L_i} \right)^2 (T_o - T_f)^2. \tag{4}$$

These last two parameterisations were used in numerous studies (e.g. review in Asay-Davis et al., 2017). As the ocean properties used to calculate melting for every draft point are taken at the very same point, they are tagged as local.

**The quadratic, local/nonlocal dependency** to thermal forcing is a new parameterisation assuming that the local circulation (at a draft point) is not only affected by local thermal forcing, but also by its average over the ice basal surface, which is written as:

$$M_+ = \gamma_T \left( \frac{\rho_{sw} c_{po}}{\rho_i L_i} \right)^2 (T_o - T_f) \langle T_o - T_f \rangle. \tag{5}$$

This formulation is inspired from Jourdain et al. (2017) that showed an overturning circulation proportional to total melt rates. It is equivalent to assuming that melting is first generated by local thermal forcing, and that this first-guess melting generates a circulation at the scale of the ice-shelf cavity that feeds back on melt rates. In other words, this formulation reflects the three equations with a uniform exchange velocity that is proportional to the cavity-average thermal forcing.

**$T_o$ and $S_o$ depth-dependence**: For these three simple functions of thermal forcing, the values of $T_o$ and $S_o$ are either depth-dependent or taken from a constant depth in the far-field (Sec. 3.3 details the different far-field ocean temperature and salinity vertical profiles). The former (for which $T_o = T_o(z)$ and $S_o = S_o(z)$) assumes a horizontal circulation between the far-field ocean and the ice-draft that would transport constant ocean properties. This can be viewed as an asymptotic case where the circulation in the cavity is driven by tides rather than melt-induced buoyancy forces. Under this assumption, this formulation

is equivalent to the aforementioned three equations with a constant and uniform velocity along the ice base.

     Alternatively, $T_o$ and $S_o$ are taken at either 500 m or 700 m depths, i.e. near the sea floor. This assumes that ocean water is advected into the cavity along the sea floor up to the grounding line, then upward along the ice base with constant ocean temperature and salinity.

     The value of $T_f$ is therefore calculated with either $S_o(z)$ in the first option, or $S_o(500)$ or $S_o(700)$ in the second option (in

a consistent way with $T_o$), but with the local ice base depth. For each far-field ocean temperature and salinity profile, we thus run three Elmer/Ice simulations for each simple function of the thermal forcing.

### 2.3.2 More complex functions of thermal forcing

The following two parameterisations attempt to improve on the above by including a representation of some of the processes that determine the temperature within the ice-ocean boundary layer. Cooling of the water as it is advected into the cavity is still

neglected, so that the waters incorporated into the boundary layer have far-field properties. However, cooling of the boundary layer by melting at depth, the rise of the waters along the ice shelf base, and the change in the freezing point with depth are all considered with different levels of detail. Critically, including such processes enables these parameterisations to simulate



**Table 2.** Physical parameters of the simulations, model grid resolutions and coupling period.

| Parameter | Symbol | Value | Unit |
|-----------|--------|-------|------|
| Ice density | $\rho_i$ | 917 | $kg\ m^{-3}$ |
| Gravitational acceleration | $g$ | 9.81 | $m\ s^{-2}$ |
| Glen's exponent | $n$ | 3 | n/a |
| Fluidity parameter | A | $6.338 \times 10^{-25}$ | $Pa^{-n}\ s^{-1}$ |
| Sea water density | $\rho_{sw}$ | 1028 | $kg\ m^{-3}$ |
| Specific heat capacity of ocean mixed layer | $c_{po}$ | 3974 | $J\ Kg^{-1}\ K^{-1}$ |
| Heat exchange velocity | $\gamma_T$ | calibrated | $m\ s^{-1}$ |
| Potential temperature of the ocean | $T_o$ | prescribed (Fig. 3) | $°C$ |
| Practical salinity of the ocean | $S_o$ | prescribed (Fig. 3) | $PSU$ |
| Latent heat of fusion of ice | $L_i$ | $3.34 \times 10^5$ | $J\ Kg^{-1}$ |
| Liquidus slope | $\lambda_1$ | -0.0575 | $°C\ PSU^{-1}$ |
| Liquidus intercept | $\lambda_2$ | 0.0832 | $°C$ |
| Liquidus pressure coefficient | $\lambda_3$ | $7.59 \times 10^{-4}$ | $°C\ m^{-1}$ |
| Elmer/Ice grid resolution | 500 m at the grounding line to 4 km away | | |
| NEMO grid resolution | 1 or 2 km in the horizontal, 10 or 20 m in the vertical (Tab. 1) | | |
| Coupling period | between 2 and 6 months (Tab. 1) | | |

regions of basal freezing, something that the simple functions of far-field temperature cannot reproduce.

**The box parameterisation** was developed by Reese et al. (2018a) based on the analytical steady-state solution of the box model of Olbers and Hellmer (2010). The latter, initially developed for a 2D cavity, represents the buoyancy-driven advection

5 of ambient ocean water into the ice-shelf cavity at depth up to the grounding line, then upward along the ice draft in consecutive boxes. The melt rates are given by:

$$BM = \gamma_T \frac{\rho_{sw}\, c_{po}}{\rho_i\, L_i} (T_k - T_{f,k}) \tag{6}$$

where the $k$ subscript indicates properties evaluated in each box. Those properties account for the transformation of ocean temperature and salinity in consecutive boxes through heat and salt turbulent exchange across the ocean boundary layer underneath

10 ice shelves. Hence, the box model is entirely driven by ocean temperature and salinity near the sea floor. Unlike plume models, the box model does not entrain deep water all along the upward transport, it advects deep water from the open ocean to the grounding zone then transports it upward. Therefore, this parameterisation produces maximum melt rates near the grounding line.





**Table 3.** parameterisations used to compute melting in stand-alone ice-sheet simulations. The last column list the calibrated $\gamma_T$ obtained from the *WARM* profile, except for the plume parameterisation where a multiplicative coefficient $\alpha$ is used instead.

| Type | Name | Information | $T_o, S_o$ | $\gamma_T \times 10^{-5}$ |
|---|---|---|---|---|
| Simple parameterisations | $M_{lin}$ | local, linear dependency to thermal forcing | depth-dependent | 2.030 |
| | $M_{lin}\_500$ | | 500 m depth | 1.060 |
| | $M_{lin}\_700$ | | 700 m depth | 0.770 |
| | $M_{quad}$ | local, quadratic dependency to thermal forcing | depth-dependent | 99.32 |
| | $M_{quad}\_500$ | | 500 m depth | 36.23 |
| | $M_{quad}\_700$ | | 700 m depth | 19.22 |
| | $M_+$ | local/nonlocal, quadratic dependency to thermal forcing | depth-dependent | 132.9 |
| | $M_+\_500$ | | 500 m depth | 36.3 |
| | $M_+\_700$ | | 700 m depth | 19.22 |
| Box parameterisation (Reese et al., 2018a) | $BM_2\_500$ | 2 boxes | 500 m depth | 2.100 |
| | $BM_2\_700$ | | 700 m depth | 1.200 |
| | $BM_5\_500$ | 5 boxes | 500 m depth | 2.240 |
| | $BM_5\_700$ | | 700 m depth | 1.250 |
| | $BM_{10}\_500$ | 10 boxes | 500 m depth | 2.840 |
| | $BM_{10}\_700$ | | 700 m depth | 1.440 |
| Plume parameterisation (Lazeroms et al., 2018) | $PME_1$ | Published implementation | Ap. D | $\alpha = 0.75$ |
| | $PME_2$ | Alternative implementation (Ap. B in the discussion paper) | Ap. D | $\alpha = 0.53$ |
| | $PME_3$ | Simple implementation | Ap. D | $\alpha = 0.32$ |
| | $PME_4$ | Asymmetric implementation | Ap. D | $\alpha = 0.63$ |

A key assumption is that the overturning circulation (i.e. volume transport through the boxes) is taken proportional to the density difference between the ambient ocean (open ocean seaward of the ice shelf) and the deepest box including an ocean-ice interface. Similarly to the simple parameterisations, the box model assumes constant heat and salt exchange velocities.

In their implementation, Reese et al. (2018a) calibrated both the heat exchange and overturning coefficients to obtain realistic melt rates for both Pine Island and Ronne-Filchner ice shelves. Here, we keep the overturning coefficient used by Reese et al. (2018a), and we calibrate the effective heat exchange velocity in the same way as the other parameterisations (Sec. 3.2).

In our implementation of the box model, the calving front position that is used to build the boxes positions is considered to be at either $x = 640\ km$ or defined by the 10 m depth contour, the limit below which no melting is permitted for the ice-sheet model. In the Reese et al. (2018a), the dependence of sub-shelf melting to the local pressure due to the vertical ice column





induces a lack of energy conservation. We thus decided not to implement this dependence, resulting in a uniform melting within each box.

For each temperature and salinity scenario, we run 6 Elmer/Ice simulations using the box parameterisation, with either 2, 5, or 10 boxes, and with ocean temperature and salinity taken at constant depths of either 500 m or 700 m.

**The plume parameterisation** developed by Lazeroms et al. (2018) emulates the 2D behaviour of the 1D plume model proposed by Jenkins (1991). This model describes the evolution of a buoyant plume originating from the grounding line with zero thickness and velocity, and temperature and salinity taken from the ambient ocean. Away from the grounding line, the thickness, velocity, temperature and salinity of the plume evolve through advection, turbulent exchange across the ocean

boundary layer underneath the ice shelf, and entrainment of deep water. Among the melt formulations presented in this paper, the plume parameterisation is the only one to include velocity-dependent heat and salt exchange velocity. No background or tidal velocity is prescribed, so turbulent exchanges and melt rates are zero right at the grounding line.

The plume model can be scaled with external parameters and applied to 1D ice drafts of any slope, ambient temperature and salinity (Jenkins, 2014). The melt rates are given by:

$$PME = M_o\,g(\theta)\,(T_o - T_{f,gl})^2\,\hat{M}(\hat{X}) \tag{7}$$

where $M_o$ is an overall scaling parameter, $g(\theta)$ is a function of physical constants (heat exchange coefficient, drag coefficient and entrainment) and ice-shelf basal slope, the $f, gl$ subscript indicates the freezing temperature at the depth of the grounding line and the final term gives the scaled melt rate, $\hat{M}$, as a universal function of scaled distance, $\hat{X}$, that was derived from empirical fitting of results generated by the full plume model on idealised geometries (Jenkins, 2014). The far-field temperature

used here is taken at the depth of the grounding line, as in the box model, and enters the parameterisation explicitly because the subsequent evolution of the ice-ocean boundary layer temperature through entrainment of the far-field ocean, melting and freezing is captured through the slope-dependent scaling and the universal function. The non-linear dependence on temperature arises because the melt rates depend on the product of plume temperature and plume speed. The latter is function of the plume buoyancy, which is itself linearly dependent on plume temperature. The physical basis for the scaling is discussed further in

Ap. B, but we note here that when the ice-shelf basal slope and far-field conditions are non-uniform, there is no longer a unique choice for those variables in the parameterisation, and choices other than the ones used in this study are equally valid.

Another major issue with the plume parameterisation is the transition from a 1D to a 2D ice draft. It is indeed difficult to identify the pathway from a given location of the ice draft to the grounding line point where the plume has emerged, which is enhanced by the fact that several plumes may end up at a given location. To define effective pathways, we use the empirical

method proposed by Lazeroms et al. (2018), which we adapted to unstructured grids (Ap. D). We also use 3 alternative methods that are described in details in Ap. D. The second method was originally proposed by Lazeroms et al. (2018) in their discussion paper but finally discarded to simplify the publication. The third method assumes that any ice-shelf point is reached by a single plume arising from the deepest grounding line point. The fourth method assumes that any ice shelf point is reached by a plume arising from the deepest grounding-line point that is found when starting from the closest grounding line point and looking for



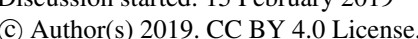



a deeper contiguous grounding-line point in the anti-clockwise direction. This fourth method is a very crude attempt to account for Coriolis asymmetry.

The plume parameterisation from Lazeroms et al. (2018) includes a heat exchange coefficient that is a function of the plume velocity along the ice-shelf base, which is similar to the ocean model but not to the other parameterisations. The complexity of

this parameterisation motivated us to calibrate it by adding a multiplicative coefficient $\alpha$ (Tab. 3) to the melt expression (Eq. 7) rather than calibrating physical parameters.

For each temperature and salinity scenario, we run 4 Elmer/Ice simulations with the plume parameterisation, using the 4 aforementioned methods to calculate the effective plume pathway and with ambient temperature and salinity taken at the effective grounding line depth (as defined in Lazeroms et al., 2018).

## 3   Experiments

### 3.1   Initial geometry and setup

We simulate the evolution of an ideal ice-sheet inspired by the Pine Island Glacier in West Antarctica. The domain is the same as the MISOMIP domain for the coupled simulations and as the MISMIP+ domain for the stand-alone ice-sheet simulations (Asay-Davis et al., 2016). The ice sheet is marine based and its grounding line rests on a retrograde bed sloping upward towards

the ocean. The entire domain, including the ice sheet and the ocean, is $800\ km$ long and $80\ km$ wide (Fig. 2). The ice-sheet calving front is located at $x = 640\ km$, while the remaining domain, up to $x = 800\ km$ and also the cavity beneath the ice shelf, is filled with ocean water. The ice sheet is in equilibrium state with an accumulation rate of $0.3\ m\ a^{-1}$ and no sub-shelf melting, as required by MISMIP+, using the ice-sheet configuration detailed in Sec. 2.1. The initial grounding line central position is $x = 450\ km$.

### 3.2   Initial state and calibration

The initial calibration purpose is to assess whether the parameterisations represent the response of melt rates to changing ocean temperature and salinity. We thus make sure that all the parameterised and coupled configurations produce the same melting average for the *WARM* profile of MISOMIP (Fig. 3, Asay-Davis et al., 2016).

We follow the ISOMIP+ protocol (Asay-Davis et al., 2016) to achieve a sub-shelf melt rate average of $30 \pm 2\ m\ a^{-1}$ below

300 m depth after 4 years of ocean spin up, with the steady ice-shelf draft shown in Fig. 3. The value of $\Gamma_T$ is not known with accuracy and usually calibrated in ocean models (Asay-Davis et al., 2016; Jourdain et al., 2017). We therefore adjust $\Gamma_T$ to achieve these melt rates (and keeping the non-dimensional salt-exchange coefficient to $\Gamma_S = \Gamma_T/35$) for each ocean-ice sheet coupled configuration (Tab. 1). Then, we compute the melting average from the sea floor up to 10 m depth from the 4 coupled configurations, which gives $\langle m_t \rangle = 8.5 \pm 1\ m\ a^{-1}$. The mean value of these melt rates is used as a target for stand-alone ice-

sheet simulations forced by the *WARM* profile from the ISOMIP+/MISOMIP protocol. For the parameterisations in which $\gamma_T$ is constant (Eq. 1), we achieve the target by adjusting $\gamma_T$. For the plume parameterisation, which accounts for a top boundary





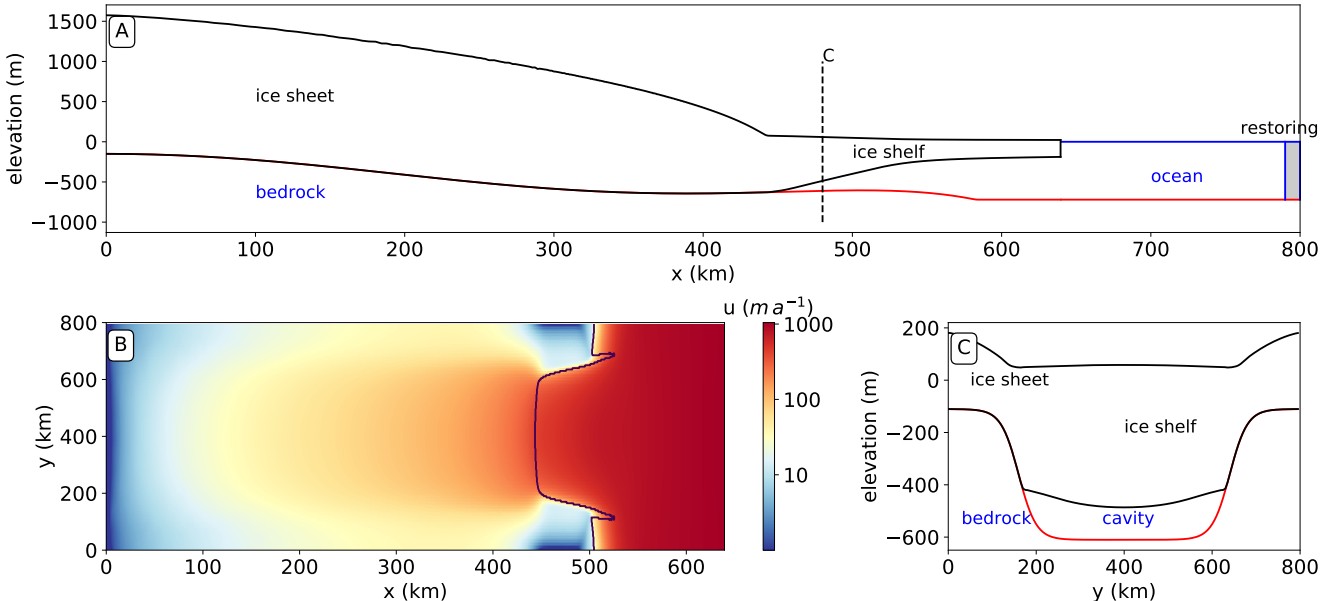

**Figure 2.** Initial ice-sheet in equilibrium calculated by Elmer/Ice with an accumulation rate of $0.3\ m\ a^{-1}$ and no sub-shelf melting as required by the MISMIP+ protocol (Asay-Davis et al., 2016). (A) Side-view geometry in the central flowline, also indicating the position of the ocean restoring used by the ocean model. (B) Velocity magnitude seen from above. The black solid line indicates the grounding line. (C) Cross section of the ice sheet at $x = 480\ km$.

layer velocity, we adjust the value of the multiplicative coefficient $\alpha$ (calibrated values shown in Tab. 3) to achieve the same target (see Sec. 2.3).

The reason why we did not calibrate the parameterisations to reproduce the average melt rates below 300 m as done in MISOMIP is because all of them produce substantial melt rates underneath the shallowest parts of the ice shelf, as opposed to the ocean models. To emphasize this point, we also performed the simulations with the calibration done as in MISOMIP below 300 m depth, the results of which are given in Ap. H.

The *WARM* profile was put forward in MISOMIP because it enables a short spin up of the ocean model, which is useful for calibration purposes as here. After this calibration phase, we keep the calibration reported in Tab. 2 for all the one-century scenarios described in Sec. 3.3.

## 3.3 The set of ocean temperature and salinity scenarios

We consider the following six scenarios over a century (Fig. 3), the first two being kept constant, and the other four linearly evolving in time:

– **Warm$_0$** resembles the present-day typical Amundsen Sea conditions (Dutrieux et al., 2014). There is no temporal change of temperature and salinity profiles.





- **Warm$_1$** starts from the *Warm$_0$* profile and then the temperature uniformly increases by 1°C/century. The salinity profile is constant in time.

- **Warm$_2$** is similar to *Warm$_1$* but the warming rate increases with depth, from zero in the surface layer to 1°C/century below the deep thermocline. The salinity profile is constant in time.

5    - **Warm$_3$** starts from the *Warm$_0$* profile and undergoes a 200 $m$ uplift of both the thermocline and the halocline.

- **Cold$_0$** resembles a cold cavity such as beneath the Ronne-Filchner ice shelves. There is no temporal change of temperature and salinity profiles.

- **Cold$_1$** starts from the *Cold$_0$* profile and then warms to reach a warm cavity state within a century. The salinity is also increased.

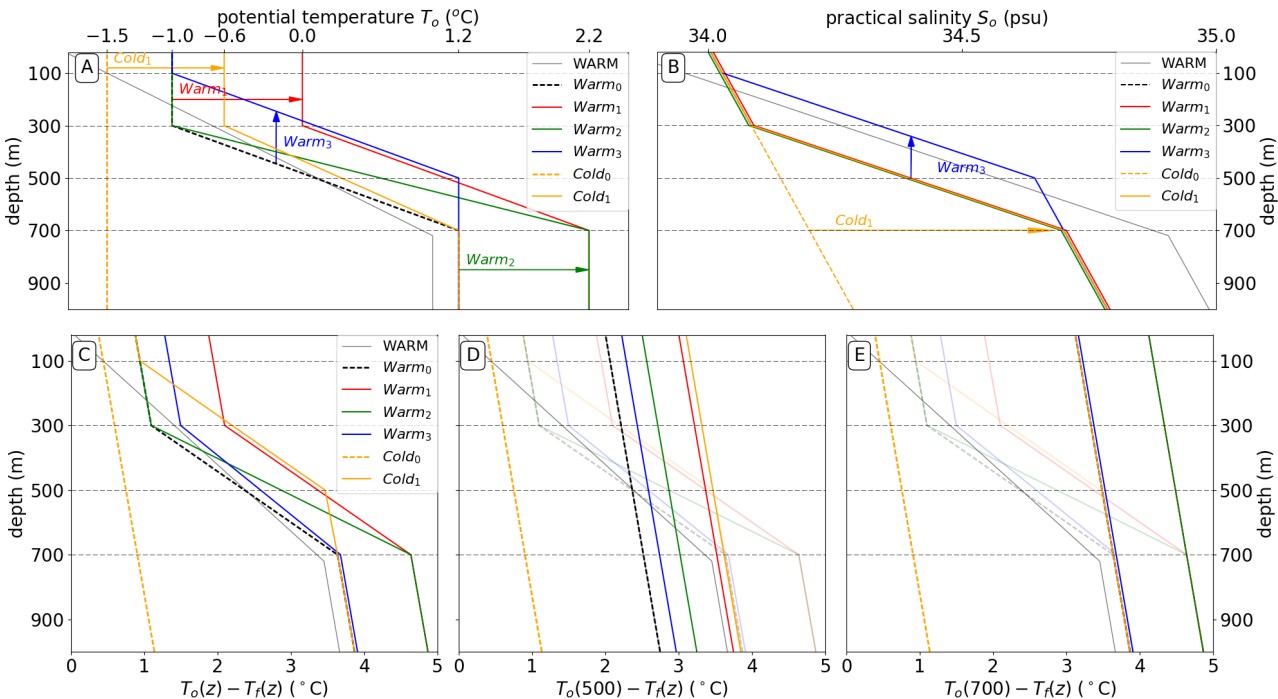

**Figure 3.** Far-field ocean temperature (A) and salinity (B) profiles scenarios in front of the cavity. The *WARM* profile is used for calibrating the initial state of parameterised and coupled simulations. The *Warm$_0$* and *Cold$_0$* scenarios are constant in time, while the others evolve linearly in time following the arrows. The *Warm$_1$*, *Warm$_2$* and *Warm$_3$* scenarios start with the *Warm$_0$* profile and end up after a century to their respective profiles, while the *Cold$_1$* scenario start from the *Cold$_0$* profile. Thermal forcing, calculated from the far-field temperature and salinity, applied to the ice-shelf draft, (D) assuming horizontal circulation between the far-field ocean and the cavity or assuming that the circulation is driven by oceanic properties at (E) 500 m and (F) 700 m depths. Profiles from the C panel are superimposed to panels D and E as a watermark for comparison purposes.





These profiles are slightly more realistic than in MISOMIP. They all include a thermocline, for previous studies have reported their importance in ice-shelf melting (e.g. De Rydt et al., 2014). The $Warm_0$ profile corresponds to a linear representation of the average hydrographic profiles measured in front of Pine Island glacier (Dutrieux et al., 2014). By contrast, the $Cold_0$ profile represents typical cold-cavity conditions in which deep ocean convection associated with sea ice formation prevents the stratification (e.g. for the Ronne and Ross ice shelves). The $Warm_1$ scenario leads to $1°$ C warming at all depths after 100 years, which corresponds to the upper 80th to 90th percentile of ocean warming projected in the Amundsen Sea by 33 CMIP5 models (Fig. G1). The $Warm_2$ scenario is more conceptual and assumes that the sea ice cover will persist over 100 years, i.e. that the ocean surface remains close to the freezing point while the subsurface gets warmer. The $Warm_3$ scenario is inspired by the study of Spence et al. (2014) suggesting that poleward shifting winds over the 21st century will uplift the coastal thermocline due to decreased Ekman downwelling. Last, the $Cold_1$ scenario is an idealized representation of the ocean tipping point described by Hellmer et al. (2012, 2017), in which the Ronne-Filchner cavities switch from a cold to a warm state.

The salinity profile is unchanged throughout $Warm_0$, $Warm_1$ and $Warm_2$ and is sufficiently stratified to keep a stable density profile. In the $Warm_3$ scenario, the halocline is lifted together with the thermocline to mimic an Ekman-driven uplift of the pycnocline, and in $Cold_1$, the stratification in salinity is increased linearly in time to keep a stable stratification when the cavity switches from cold to warm states.

The bottom panels in Fig. 3 show the thermal forcings applied to stand-alone ice-sheet simulations for the different hypotheses for temperature and salinity inputs (Sec. 2.3), while Tab. 3 summarises the ensemble of sub-shelf melting parameterisations.

## 4   Results

### 4.1   Melting patterns resulting from the initial calibration

The calibrated parameters are given in Tab. 3 and the melting patterns are shown in Fig. 4 (not all the patterns are shown). The patterns obtained from the coupled and parameterised simulations are quite different, even though all of them result in similar cavity melt rates. The coupled simulations give most melting below approximately 300 m depth and almost no melting near the ocean surface, which also highlights why the calibration was performed below 300 m depth in ISOMIP+ (Asay-Davis et al., 2016). The parameterised simulations give significant melt rates at all depths.

Near the grounding line, melt rates higher than $50\ m\ a^{-1}$ are predicted by all coupled simulations, while this value is only and hardly reached by the $M_{quad}$ parameterisation and never reached in the other cases. Away from the grounding line, where the ice shelf is also thinner, melt rates are close to zero for the coupled simulations while they mostly remain above $10\ m\ a^{-1}$ when parameterised. Such differences in melt rate patterns are expected to induce diverging responses from the ice sheet (Gagliardini et al., 2010; Reese et al., 2018b).

While the patterns in the coupled simulations are quite similar to each other, the parameterised patterns differ to various extents. The parameterisations having a simple dependence to thermal forcing (i.e. $M_{lin}$, $M_{quad}$ and $M_+$) compute the highest melt rates at depth, which also falls close to the grounding line in the central flowline. They also result in a rather uniform pattern when the basal surface is closer to the sea surface, which falls away from the grounding line in the central flowline but



**Figure 4.** Diagnostic sub-shelf melt rates obtained through the calibration process by forcing the coupled and the parameterised models with the *WARM* profile from Asay-Davis et al. (2016). All the ocean members are represented (last column) but not all the parameterisations (first three columns). The average melting for every parameterisation equals $8.5\ m\ a^{-1}$, while being in the range $8.5 \pm 1\ m\ a^{-1}$ for the ocean members. In the $PME_1$ panel are shown the 200 m, 300 m and 400 m depth contours. The grounded ice is coloured in grey.

also close to the grounding line on the sides of the ice shelf, where two bits (or horns) of grounded ice penetrate seaward. The range of melt rates is wider for the $M_{quad}$ parameterisation, thinner ice being less melted and thicker ice being more melted, compared to $M_{lin}$ and $M_+$. The $M_{lin}$ and $M_+$ patterns are identical by construction because the melting average is driven by the $(T_o - T_f)$ term, which appears only once in the two respective formulations. However, the respective calibrations are





different (Tab. 3) because of the term $\langle T_o - T_f \rangle$ appearing in $M_+$ only, and the sensitivity to ocean warming will therefore be different.

The implementations of the 2D plume emulator produce quite different patterns between $PME_1$, $PME_2$, $PME_3$ on the one hand and $PME_4$ on the other hand, mostly because the latter is highly asymmetric. In the first three implementations, the different approaches adopted to calculate the effective depth and angle (Lazeroms et al., 2018) all result in very similar patterns. They all induce zero to small melt rates near the central grounding line because there the plumes arise from a limited number of directions. However, along the sides of the main trunk, on the inner side of the horns, the melt rates gets higher at the grounding line because the plumes can emerge from more deeper portions. Farther away, near the calving front, many more plumes can be conbined (for $PME_1$ and $PME_2$) and contribute to a increasing melting towards the ice front. These high melt rates near the calving front also reflect the empirical scaling made in Lazeroms et al. (2018) to link them to the distance to the grounding line, which may not be adapted to our relatively small ice shelf. In the $PME_3$ parameterisation, the plume can only come from the deepest grounding line, which is also crossing the central flowline, whatever the position in the ice draft. On the external sides of the domain, it induces strong melting compared to $PME_1$ and $PME_2$ for which the plumes can also come from less deep parts of the cavity and mitigate the melt rates.

Similarly to the $M_{lin}$, $M_{quad}$ and $M_+$ parameterisations, the box parameterisation produces its highest melt rates near the grounding line. Away from the grounding line, the melt rates get lower to end up with the lowest values close to the calving front. The larger the number of boxes, the larger the melt rates near the grounding line, and the smaller the melt rates near the calving front.

### 4.2 Ice mass loss and sub-shelf melt rates

The initial ice sheet is built within the framework of MISMIP+ (Asay-Davis et al., 2016) requiring no sub-shelf melting, and is thus in equilibrium under such conditions. The simulations thus all start with an initial dynamical adjustment of the ice-sheet geometry to new ocean conditions (Fig. 4), which generates a melting pulse despite the 5 years of ocean spin-up. The adjustment is larger for relatively warmer scenarios (Fig. 5). The pulse is therefore much lower and hardly visible for the $Cold_i$ scenarios. For the $Warm_i$ scenarios, the peak of the pulse yields similar melting of up to 130 $Gt\ a^{-1}$ for parameterised and coupled simulations. However, it lasts longer for the latter, about 20 a, than for the former, about 5 a.

The initial pulse is followed by a melting minimum of about 50-60 $Gt\ a^{-1}$ for both the coupled and parameterised simulations when forced by the $Warm_i$ scenarios, and of about 20 $Gt\ a^{-1}$ for the $Cold_i$ scenarios. Then apart from $Cold_0$, all scenarios lead to further melting increase, which ends up with between 40 and 175 $Gt\ a^{-1}$ for the $Warm_i$ scenarios and between 60 and 90 $Gt\ a^{-1}$ for $Cold_1$. This makes the ice-sheet contributing 4 to 12 mm to sea level for the $Warm_i$ scenarios, 2 to 4 mm for the $Cold_1$ scenario and 0.5 to 3 mm for the $Cold_0$ scenario (Fig. 6).

For the $Warm_i$ scenarios, the parameterisations in general tend to overestimate the melting close to the surface and underestimate it at depth. This results in initially melting a large part of thinner ice, which makes overall melting higher compared to coupled simulations. Along with the disappearance of thinner ice, the overall melting becomes progressively lower than for coupled simulations. In the end, this results in lower sea level contribution (SLC) from the parameterised simulations, apart



**Figure 5.** Same as Fig. 6 but showing the total melt rates for all experiments. The coupled simulations are shown in solid light grey but their envelope is not shown. The coloured lines correspond to parameterised simulations. The black solid lines correspond to a 50% underestimation/overestimation compared to the average of coupled runs members.





**Figure 6.** Sea level contribution (SLC) for all the experiments. The coupled simulations are shown in solid light grey and their envelope in grey shading. The coloured lines correspond to parameterised simulations. The black solid lines correspond to a 50% underestimation/overestimation compared to the average of coupled runs members.





from few exceptions. In the $Cold_i$ scenarios, melting is never high enough to completely remove thin ice and the SLC from parameterised simulations is more in agreement with the coupled simulation on average.

The uncertainties linked to the ocean model are emphasised by the spread of SLC calculated from the coupled model. The spread is about $\pm 10\%$ around the average for all the scenarios but the $Cold_0$ and $Warm_2$ scenarios where it is about $\pm 20\%$,
respectively. These uncertainties are induced by melt rates, which results from the ocean model response to far-field ocean forcing, but also by the different responses of the ice sheet to these melt rates. Neither of them can be discarded but the fact that the spread is higher for the $Warm_2$ scenario may reflect a higher range of sensitivities of the coupled system to warming at depth, compared to the other warming scenarios. A larger spread of about $\pm 30\%$ for the $Warm_i$ scenarios, and about $\pm 50\%$ and $\pm 100\%$ for the $Cold_1$ and $Cold_0$ scenarios, respectively, is obtained from the parameterisations, which reflects the wide
variety of approaches and indicates that it makes sense to inter-compare parameterizations with respect to the coupled model.

Whatever the type of hypothesis for the depth at which the far-field ocean temperature and salinity profiles are taken (Sec. 2.3), the $M_{lin}$ parameterisations tend to largely overestimate the melt rates for the $Cold_i$ scenarios, and underestimate them for the $Warm_i$ scenarios, leading to respectives overestimation and underestimation of SLC. This reflects a poor representation of melting by these parameterisations when the change in ocean forcing is too large.

The $M_{quad}$ parameterisations give melting in fair agreement with coupled results for the $Cold_i$ scenarios. For the $Warm_i$ scenarios, the tendency is a slight underestimation of SLC using the $M_{quad}$ and $M_{quad\_}700$ parameterisations, and a larger underestimation using $M_{quad\_}500$. Compared to the $M_{lin}$ parameterisations, it behaves much better and for a larger range of scenarios. All the $M_{quad}$ parameterisations behave quite well when confronted to a rise in the thermocline ($Warm_3$ scenario).

The $M_+$ parameterisation results are almost as close to the coupled simulations as the $M_{quad}$ parameterisations for the
$Cold_i$ scenarios, and closest for the $Warm_i$ scenarios. Regarding all the scenarios, this makes this parameterisation the best among simple parameterisations. When the far-field ocean temperature and salinity profiles are taken at depth, the results are comparable to the $M_{quad\_}500$ and $M_{quad\_}700$ parameterisations, thus slightly underestimating SLC.

Forcing a parameterisation by the far-field depth-dependent or the constant depth ocean properties changes the thermal forcing at the ice-ocean interface (Fig. 3) but also the initial calibration (Tab. 3). Considering a constant depth for instance, the
deeper the depth, the larger the thermal forcing, but also the lower the calibrated parameter, which affects the further evolution of melt rates in a complicated way. For instance, the simple parameterisations forced by constant depth ocean properties result in less SLC, which also reflects the increase in thermal forcing compared to the depth-dependent forcing. However, if the given constant depth is 700 m, the thermal forcing is larger compared to 500 m depth but the SLC is lower. Such situation also happens for the $Warm_i$ ($i = 0, 1, 2$) scenarios and is due to the initial calibration, but could also be an illustration that less
melting can result in more ice loss depending on the exact melting patterns (Gagliardini et al., 2010).

The quality of the $PME_i$ parameterisations results, in regard to the coupled simulations, is linked to the degree of warming. The higher the thermal forcing, the poorer are the results. The SLC is systematically underestimated except for the coldest ($Cold_0$) scenario for which the SLC prediction is in agreement with the coupled results. In terms of melt rates, this parameterisation computes a different pattern compared to the other parameterisations. The melt rates are very low near the central
grounding line and almost uniform downstream. This could explain why, compared to the other parameterisations, the prior



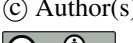

pulse that they undergo is shorter in time and why after this pulse the melt rates drop down to much lower melt rates compared to others. After this pulse, the ice-shelf is mostly composed of thick ice, and the low melt rates near the grounding line, where the ice is thicker, hamper the impact of melting on buttressing relatively to the coupled and parameterised simulations. Surprisingly, the $PME_i$ parameterisations are quite close to one another, regardless of the approach used to define effective grounding line and angle.

The box parameterisations are forced by the ocean properties at a constant depth, being either 500 m or 700 m depths. Whatever the depth, the higher the number of boxes, the larger both the overall melting and the SLC in our experiments, which is enhanced for the $Warm_i$ scenarios compared to the $Cold_i$ scenarios. The optimal number of boxes for the $Cold_i$ scenarios is between 2 and 5, while for the $Warm_i$ scenarios using 5 boxes results in a good agreement with the coupled simulations and seems to be the best trade-off within the box model, regardless of the given forcing depth. Note that using 700 m for the forcing depth gives pretty good results whatever the number of boxes, while using 500 m ends up in a larger spread in our experiments.

A rise of the thermocline ($Warm_3$ scenario) does not affect the coupled simulations, likely because sea-floor ocean properties remain unchanged in this experiment. This emphasises the importance of sea-floor ocean properties for ice-shelf melting, and explains why the box model is closer to the coupled model when ocean properties are taken at 700 m depth.

## 5    Discussion

Parameterising sub-shelf melt rates in ice-sheet modelling is currently the only way to account for melting in large-ensemble or multi-millenium simulations of the Antarctic ice sheet (DeConto and Pollard, 2016), and even shorter term simulations applied to single Antarctic basins have been done at very few occasions and only very recently (Thoma et al., 2015; Seroussi et al., 2017). Our study suggests that parameterisations should be chosen with caution. To assess the capacity of the parameterisations to reproduce the ocean-induced melting and its effect on ice-sheet dynamics under a wide range of scenarios, we setup a performance indicator (Fig. 7). We define it as the root-mean-square deviation (RMSD) in SLC of every parameterisation with respect to the average of coupled simulation on a given year. We choose to calculate this performance indicator at the fiftieth year of the simulations, for a significant part of the ice shelf is melted out by the parameterisations after this year.

While the plume parameterisation is in pretty good agreement with coupled simulations for the cold forcings, it consistently underestimates both the melt rates and subsequent SLC for the warm forcings. Lazeroms et al. (2018) show melt rates patterns in good agreement with observations for the large ice shelves such as Ronne-Filchner and Ross. However, for smaller ice shelves such as the Pine Island and Thwaites glaciers, the patterns exhibit very strong melting near the calving front, and quite a uniform melting in the entire cavity. This is contradictory to observation-based estimates (Rignot et al., 2013a; Dutrieux et al., 2013) and to high-resolution ocean simulations (Dutrieux et al., 2014) showing large melt rates near the grounding line that drop abruptly a few kilometres downstream to almost zero near the calving front. In our coupled model configuration, the melt rates are zero at the grounding line, close to zero nearby (not seen in the Lazeroms et al. (2018) paper because of too coarse resolution) and the strongest at the calving front. We suspect this is due to the empirical relationship used in Lazeroms et al.



**Figure 7.** Performance of parameterisations compared to coupled simulations, calculated at the fiftieth year of simulations. (A) Root mean square deviation (RMSD) in SLC of every parameterised simulation with respect to the average of coupled simulations. (B) Difference between SLCs from parameterisations and coupled simulations for all the experiments. The grey shading is only to ease the comparisons between the parameterisations.

(2018) that relates melt rates to the depth difference between the effective grounding line point and the ice draft, which may wrongly place the melting-accretion point for small ice shelves as opposed to large ice shelves. The fact that the same ice-sheet response occurs regardless the type of implementation supports this point.

The box parameterisation tends to give relatively good results regardless of the number of boxes or the near sea-floor depth at which the ocean properties are taken. Using 5 boxes seems to yield the best results. Reese et al. (2018a) found that increasing





the number of boxes in a static cavity would converge to almost constant average melt rates above 5 boxes. In our study, increasing the number of boxes neither lead to convergence of the calibrated parameter, nor to converging SLC during the prognostic simulations. The melting pattern has an effect on the ice sheet dynamics, so even though convergence could be expected from the work of Reese et al. (2018a) for a static cavity, the ice sheet response to the different patterns related to the various number of boxes could have suppressed the initial convergence.

A key issue in our implementations of the 1D plume parameterisation might be in the use of deep ocean temperatures, which will lead to an overestimate of melting near the ice front. Our calibration procedure then scales back the melting near the grounding line and leads to an underestimate of the reduction in buttressing. The box model also uses the deep temperatures, but in that parameterisation heat is supplied to the overturning circulation in the grounding zone only, beyond which melt rates must fall as a result of the extraction of latent heat and the rise in the freezing point. Calibrating the heat transfer coefficient alters the balance between heat used to melt in the grounding zone and that advected downstream to melt elsewhere. Hence, the calibration redistributes the melting rather than just scaling a fixed melt pattern, and that may be the reason that the results compare quite well with those from the coupled model, especially when the parameterisation is used with five boxes.

Among the simple functions of thermal forcing, the two quadratic, local and nonlocal, functions are in good agreement with the coupled simulations. A nonlocal dependency leads to slighlty better results. Taking the ocean properties at a varying depth gives better results. In that case, these two parameterisations are the only ones to capture the increased melting of coupled simulations after the initial adjustment phase in the $Warm_1$ scenario. When these simple functions depend on constant depth ocean properties, deeper temperature and salinity inputs results in better agreement with coupled simulations.

We chose to calibrate the parameterisations using the same far-field ocean temperature and salinity constant profiles, which is different from the temperature and salinity scenarios used in the rest of the study. Such approach is actually very selective but enables to distinguish between parameterisations that could be applied to real cases, because they adapt well to a change in ocean properties, from those that either need to be improved or discarded in regard to changing ocean conditions.

All parameterisations yield too large melt rates in thin ice areas and too small melt rates near the deepest parts around the grounding line. Even though our geometrical setup is ideal, the distribution of thicknesses within the ice shelf are not far from reality, meaning that applying these parameterisations to real ice shelves would also induce too much thinning of initially thin floating ice. The studies of Jenkins (2016) and Jenkins et al. (2018) suggest that the basal slope of the ice shelf influences the mixing across the thermocline. Accounting for this effect in simple functions of thermal forcing may allow to redistribute more melting over the steep areas near grounding line and less melting over flat areas near calving fronts, thus decreasing the overmelting of thin floating ice.

The choice of a parameterisation for real applications may account for the local circulation in the ice-shelf cavity. Whether the circulation is horizontal or vertical may guide the choice of the dependence to thermal forcing being either a function of varying depth or taken at a constant depth. For instance, the circulation in the Amundsen sea embayment appears to be a mix between vertical overturning fed by incursions of CDW and horizontal barotropic flow generated by tides (Jourdain et al., 2017, 2018). It should be noted that our study does not account for sea ice, which tends to limit the ekmann pumping due to wind stress and vertical mixing, nor for tides.



The spatial distribution of melt rates affect ice-shelf buttressing in a complicated way. Similar total melt rates distributed differently beneath the ice shelf is likely to induce distinct responses of the ice sheet (Reese et al., 2018b; Gagliardini et al., 2010). Conversely, different melting patterns can induce similar responses of the ice sheet if the integrated loss in buttressing happens to be well balanced from one another. This is illustrated in our simulations, for instance by the two types of quadratic

functions of the thermal forcing that exhibit different patterns but lead to similar SLC. The study of Reese et al. (2018b) attributes an equal effect of bits of ice shelf removal on ice-sheet dynamics in places where ice thicknesses can be very different. Removing floating ice near the deepest grounding lines or near ice rises can remove the same amount of buttressing and lead to similar SLC. Ice rises are generally found in shallow waters, thus a parameterisation that computes too large melt rates near this sensitive area may remove too much buttressing restraining the upstream ice sheet compared to coupled simulations.

An ocean-ice sheet coupled model is needed as a reference to assess the melting parameterisations. Only an ocean model can convey the complexity of ocean physics to melting at the ice-shelf base, as opposed to parameterisations, and only an ice-sheet model can respond to a change in ice-shelf buttressing induced by changing melt rates. On the one hand, the ocean model NEMO was used to calculate the melt rates in the coupled framework. On the other hand, the ice-sheet was simulated by the Elmer/Ice model using the SSA* approximation of the Stokes equations and a Schoof friction law at the ice-bed interface. Over

the last decade, many ice-sheet and ocean models were developed, which motivated various model intercomparison projects to evaluate the caveats and assets of models and their physics in regard to ideal simulations (The MISMIP and MISMIP3D projects in Pattyn et al. (2012) and Pattyn et al. (2013) for ice sheet models, the ISOMIP project in Holland et al. (2003) for ice shelf-ocean models and the MISMIP+, ISOMIP+ and MISOMIP1 projects in Asay-Davis et al. (2016) for ice sheet, ice shelf ocean and ocean-ice sheet coupled models). These intercomparison projects have highlighted differences between

models that have not been accounted for in our study, even though we included an ensemble of coupled configurations to quantify uncertainties in the ocean model grid and physics. Pursuing this present study using other types of models and physics will be needed to further assess the robustness of our results.

Our study highlights the assets and caveat of sub-shelf melt parameterisations that can be constrained by the far-field ocean, some of which being used over a decade without thorough assessment. This work was performed with an idealised representa-

tion of a relatively small outlet glacier in West Antarctica, and now needs to be extended to Antarctic realistic ocean-ice sheet systems in order to improve sea level projections.

## 6   Conclusions

A wide variety of sub-shelf melting parameterisations depending on oceanic properties has been compared to an ensemble of ocean-ice sheet coupled simulations. A new coupled model combining the ocean model NEMO and the ice-sheet model

Elmer/Ice has been presented. Among the complex parameterisations that we assessed, representing melting through a 2D emulation of a 1D plume model gives good results for cold scenarios but underestimates the melt rates and sea level contribution for warm scenarios. Given the high degree of complexity in the physics represented in the plume model, it is possible that calibrating more parameters could improve the validity of the scaling across multpiple ice-shelf sizes. More work may also





improve the way to extend the 1D plume model to a realistic ice draft. The box parameterisation representing the vertical overturning in the cavity gives results relatively close to the coupled simulations, especially when used with five boxes. We have shown that a linear parameterisation of thermal forcing is not able to represent ocean induced melting beneath an ice shelf. Instead, a quadratic parameterisation of thermal forcing gives much better results, which are even improved for a local/nonlocal

5   approach, as opposed to a fully local approach. Studies aiming at projecting the future contribution of Antarctica to sea level should care about the choice of the melting parameterisation before providing predictions. We recommend to validate the chosen parameterisation in regard to ocean-ice sheet model coupled simulations within each specific environmental conditions and ice physics, although our results have to be taken carefully until assessment based upon other models are produced.

*Code availability.*   We used Elmer/Ice Version 8.3 at revision 6be9699, which is available at git://www.github.com/ElmerCSC/elmerfem, and

10   NEMO-3.6 at revision 6402. The experimental protocol is composed of: the coupling framework version 1.1, available at http://zenodo.org/badge/latestdoi/; the NEMO setup version Feb-2019, available at http://doi.org/10.5281/zenodo.2562731; and the Elmer/Ice setup version 1.2, available at http://doi.org/10.5281/zenodo.2563156.

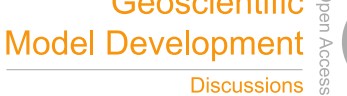



## Appendix A: Schoof friction law

The Schoof friction law is written as in Brondex et al. (2017) and Brondex et al. (2018) as follows:

$$\tau_b = \frac{C_s\,u_m^b}{\left(1 + \left(\frac{C_s}{C_{max}\,N}\right)^{1/m}u_b\right)^m} \tag{A1}$$

The parameters are explained in Tab. A1.

**Table A1.** Parameters of the Schoof friction law.

| Parameter | Symbol | Value | Unit |
|---|---|---|---|
| Basal friction exponent | $m$ | 1/3 | n/a |
| Fluidity parameter | A | $6.338 \times 10^{-25}$ | $Pa^{-n}\,s^{-1}$ |
| Friction parameter | $C_s$ | $3.16 \times 10^6$ | $Pa\,m^{-m}\,s^m$ |
| | $C_{max}$ | 0.5 | n/a |

## 5  Appendix B: Sensitivity to the coupling period

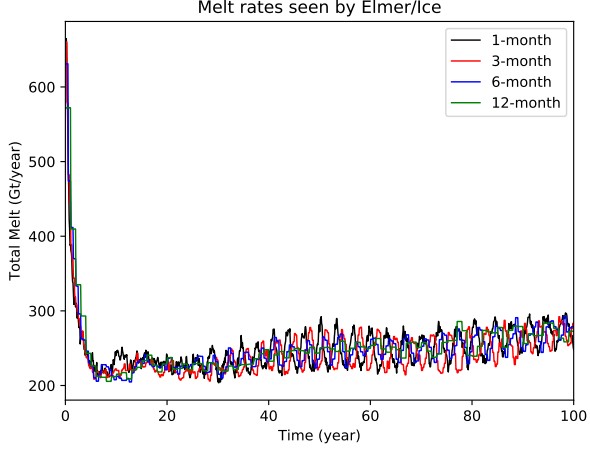
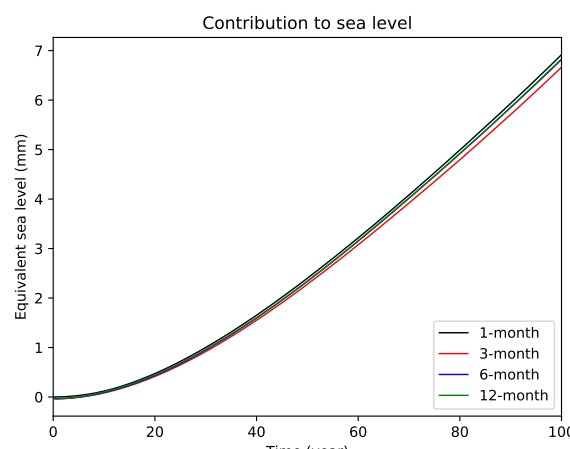

**Figure A1.** Mean cavity melt rate seen by Elmer/Ice for various coupling periods (left panel). Global mean sea level rise equivalent to the ice mass loss simulated by Elmer/Ice for various coupling periods (right panel). The four simulations correspond to the COM-Ocean1 experiment of the standard MISOMIP protocol (Asay-Davis et al., 2016).

.




## Appendix B: Physical basis for the plume parameterisation empirical scaling

The plume parameterisation is derived empirically from the results of a full plume model (Jenkins, 2014) applied to a range of simple ice shelf geometries and water properties. If the ice-shelf base is linear and the far-field ocean uniform, results for a wide range of ocean temperatures, ice-shelf basal slopes and grounding-line depths, when appropriately scaled, collapse

(within ±20%) onto a universal melt rate curve (Jenkins, 2014). The plume parameterisation of Lazeroms et al. (2018) was created by fitting an 11th order polynomial function to the universal curve.

When applying the parameterisation in practice, there are a number of issues to deal with: the ice-shelf basal slope will vary; the far-field ocean will be non-uniform; and for 2D ice-shelf geometries, there is no unique grounding-line point. The first two are generic problems that arise from the simplifications that are required to allow the derivation of a universal melt rate curve.

The latter arises when the 1D parameterisation is implemented in 2D.

The ice shelf basal slope enters the parameterisation through the function:

$$g(\theta) = \left( \frac{sin\theta}{C_d E_0 sin\theta} \right)^{1/2} \left( \frac{C_d^{1/2} \Gamma_{TS}}{C_d^{1/2} \Gamma_{TS} + E_0 sin\theta} \right)^{1/2} \left( \frac{E_0 sin\theta}{C_d^{1/2} \Gamma_{TS} + E_0 sin\theta} \right) \tag{B1}$$

The last of these terms scales the thermal driving in the plume as a fraction of the far-field thermal driving, while the first two scale the plume speed based on the balance between buoyancy and friction (first) and the dependence of the buoyancy on

far-field thermal driving (second). Since the inertia of the plume is small, its speed rapidly adjusts to changing slope, and the first term of the above expression therefore represents a local balance between the upslope buoyancy force and frictional drag. The latter two terms, on the other hand, reflect the balance between entrainment and melting over the path of the plume, so cannot be directly related to the local slope, if the slope is non-uniform. However, for low slopes the turbulent transfer of heat and momentum at the ice base tend to dominate over entrainment, giving:

$$g(\theta) = \left( \frac{sin\theta}{C_d} \right)^{1/2} \left( \frac{E_0 sin\theta}{C_d \Gamma_{TS}} \right) \tag{B2}$$

Hence, for low slopes the thermal driving evolves along the plume path with a simple $sin\theta$ scaling that effectively makes it a function of the depth change between the grounding line and the point of interest. It does not matter if that path is short and steep with rapid entrainment, or long and gentle with slow entrainment, the net result is the same. The first term remains a local scaling, so when the parameterisation is applied to 1D problems with varying slope, using the local slope to estimate $g(\theta)$

gives good results (Lazeroms et al., 2018).

An equivalent solution to the problem of non-uniform far-field properties is less obvious. Taking a depth-average to reflect the range of properties entrained into the plume is the option used in Lazeroms et al. (2018). However, entrainment is strongest where the basal slope is steepest, so we might expect deeper waters to contribute more to the plume. In this study, we use the temperature at the depth of the grounding line. Since the temperature profiles all have a thick isothermal layer at the seabed,

taking an average over the depth range from which waters are entrained into the plume would probably yield similar results, because the isothermal layer would dominate.





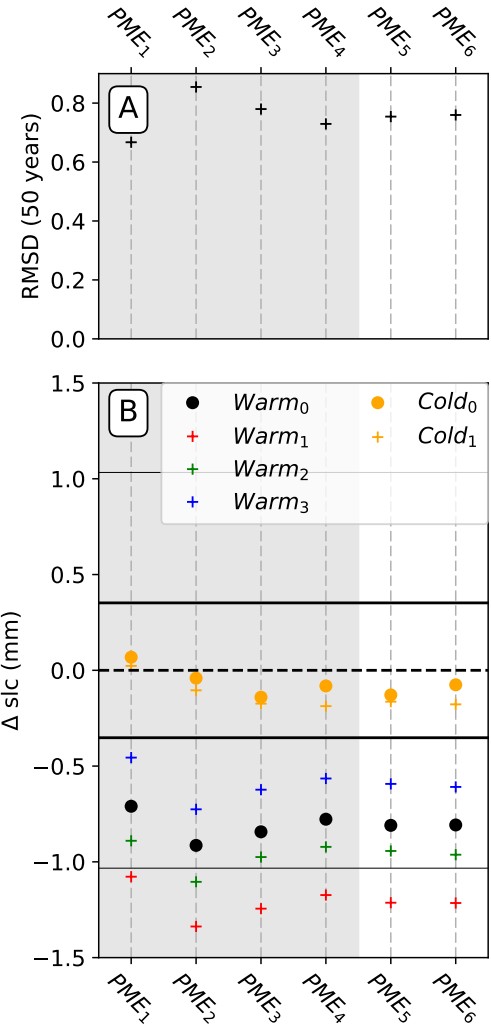

**Figure C1.** Similar to Fig. 7 to evaluate the use of the local gradient in $PME_3$ (which gives $PME_5$) and $PME_4$ (which gives $PME_5$) to calculate the effective angle instead of using the slope between the ice draft and the grounding line from which starts the plume. $\alpha = 0.34$ and $\alpha = 0.65$ for the calibration of $PME_5$ and $PME_6$, respectively (Sec. 3.2). The grey shading is only to ease the comparisons between the parameterisations.

Implementation of the plume parameterisation in 2D is a more complex problem, to which there are many possible solutions. The procedure implemented by Lazeroms et al. (2018) was effectively an average of 1D implementations along whichever of 16 prescribed directions represented valid plume paths. For each valid direction the grounding-line depth and the local slope in that direction were used to scale the melt rates. In $PME_1$ we implemented that procedure as closely as possible, given the 5 unstructured model grid (Ap. D). However, following the above reasoning, it could be argued that the path followed by the





plume should not matter. Only the depth change from the grounding line to the point of interest should influence the results and the plume should flow locally up the steepest slope, i.e. parallel to the gradient vector. Using the magnitude of the local gradient vector as the slope scale ($PME_2$) suppresses the checkerboard noise in the melt rates, but does not greatly influence the results. In $PME_3$ and $PME_4$, we adopted two procedures for picking a unique grounding line point for each grid point, rather than using an average of many. In each case, we scaled the melt rate using the depth of the grounding-line point and the mean slope along a straight line connecting the grounding line point and the grid point. Following the earlier reasoning, using the magnitude of the local gradient vector is equally valid. Results using that alternative are shown in Fig. C1, but differ little from those presented in Fig. 6.



## Appendix D: Implementations of the plume parameterisation in Elmer/Ice

The plume parameterisation was originally implemented for regular grids. We adapt the method used to calculate the effective grounding line depth and effective angle as defined in Lazeroms et al. (2018) to the unstructured grids used in Elmer/Ice. Here, we describe the adapted method and the alternative implementations also discussed in the paper.

### D1   $PME_1$

In the original algorithm published in Lazeroms et al. (2018), the melt rates at a draft point are calculated by considering the effective grounding line depth, which is calculated by searching in the 16 grid directions equally distributed around the draft point and starting from it, insofar as those directions are valid. The 16 directions follow the grid points as shown in Fig. 3 of Lazeroms et al. (2018). A direction is valid if (i) the local slope in this direction is negative and (ii) the first grounded point met in this direction is deeper than the draft point in this direction. The effective grounding-line depth is calculated using Eq. 13a of Lazeroms et al. (2018), and the effective angle is calculated using Eq. 13b by considering the local slopes in the valid directions.

Instead of grid directions, we consider directional triangles that are angularly equally distributed around the draft point. Anologolously to the original criterion to find valid directions, the criterion to make a cone valid is based on (i) the average of the local angles of all the directions connecting the draft point to the grounding line points included in the cone and (ii) the average of these grounding line points depths (Fig. E1A). The simulations were all done using 64 triangles around the draft point, which enables a rather smooth melting pattern compared to using 16 triangles (analogously to Lazeroms et al. (2018) for directions).

### D2   $PME_2$

In the algorithm published in Ap. 2 of Lazeroms et al. (2017), i.e. the discussion version of Lazeroms et al. (2018), the criterion to make a direction valid is the same as the first algorithm, but the computation of the effective grounding-line depth and angle is slightly different, taking for instance the local gradient instead of the local slope in each direction to calculate the effective angle.

We calculate the effective grounding line depth and angle as it is in Eq. B1 and B2 of Lazeroms et al. (2017) but using the search for valid directions as explained in Sec. D1 (Fig. E1A), also using 64 triangles.

### D3   $PME_3$

In this algorithm, we simply take the deepest grounding line (which is located in the central flowline) to calculate the effective grounding line depth, and the effective angle is the slope between this grounding-line point and the draft point (Fig. E1B).





### D4  $PME_4$

This algorithm accounts for the asymmetry resulting from the Coriolis effect, although in a very crude manner. The effective grounding-line depth is found by starting from the closest grounding-line point and looking for the a deeper contiguous grounding-line point in the anto-clockwise direction until as long as the grounding line deepens. Two examples of this algorithm are shown in (Fig. E1C) for two draft points in the left and the right side of the cavity, respectively. The effective angle is calculated as in D3.

**Figure E1.** Computing the effective grounding-line depth and angle for the four plume parameterisations implementations. Panel A illustrates both the published implementation $PME_1$ (Lazeroms et al., 2018) and the one appearing in the corresponding discussion article $PME_2$ (Lazeroms et al., 2017), here with 12 directions (vs 64 used in the paper), panel B illustrates the simple implementation $PME_3$ and panel C illustrates the asymmetric implementation $PME_4$.



### E1  $PME_5$

This implementation is similar to $PME_3$ but the effective angle is calculated from the ice-draft local gradient. The results are not given in the main article but compared to the other plume parameterizations in Fig. C1.

### E2  $PME_6$

5   This implementation is similar to $PME_4$ but the effective angle is calculated from the ice-draft local gradient. The results are not given in the main article but compared to the other plume parameterizations in Fig. C1.





## Appendix F:  CMIP5 temperature anomalies in the Amundsen Sea under the RCP8.5 scenario

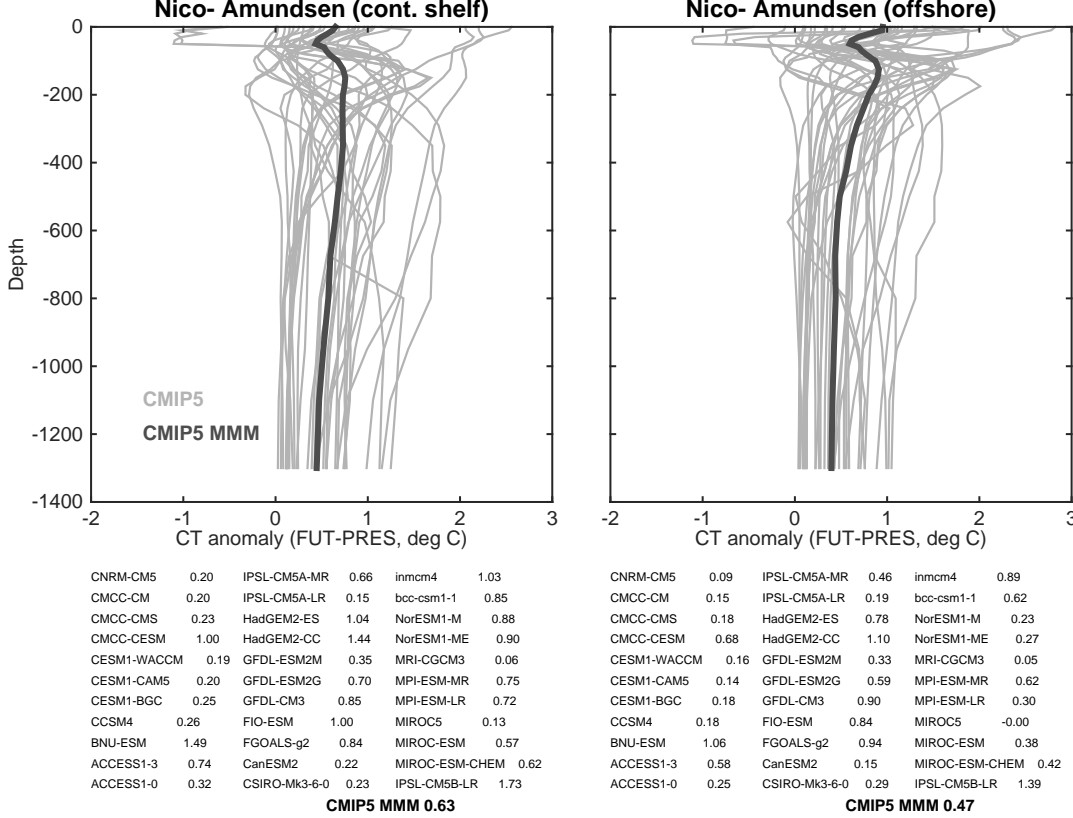

| CNRM-CM5 | 0.20 | IPSL-CM5A-MR | 0.66 | inmcm4 | 1.03 |
|---|---|---|---|---|---|
| CMCC-CM | 0.20 | IPSL-CM5A-LR | 0.15 | bcc-csm1-1 | 0.85 |
| CMCC-CMS | 0.23 | HadGEM2-ES | 1.04 | NorESM1-M | 0.88 |
| CMCC-CESM | 1.00 | HadGEM2-CC | 1.44 | NorESM1-ME | 0.90 |
| CESM1-WACCM | 0.19 | GFDL-ESM2M | 0.35 | MRI-CGCM3 | 0.06 |
| CESM1-CAM5 | 0.20 | GFDL-ESM2G | 0.70 | MPI-ESM-MR | 0.75 |
| CESM1-BGC | 0.25 | GFDL-CM3 | 0.85 | MPI-ESM-LR | 0.72 |
| CCSM4 | 0.26 | FIO-ESM | 1.00 | MIROC5 | 0.13 |
| BNU-ESM | 1.49 | FGOALS-g2 | 0.84 | MIROC-ESM | 0.57 |
| ACCESS1-3 | 0.74 | CanESM2 | 0.22 | MIROC-ESM-CHEM | 0.62 |
| ACCESS1-0 | 0.32 | CSIRO-Mk3-6-0 | 0.23 | IPSL-CM5B-LR | 1.73 |

**CMIP5 MMM 0.63**

| CNRM-CM5 | 0.09 | IPSL-CM5A-MR | 0.46 | inmcm4 | 0.89 |
|---|---|---|---|---|---|
| CMCC-CM | 0.15 | IPSL-CM5A-LR | 0.19 | bcc-csm1-1 | 0.62 |
| CMCC-CMS | 0.18 | HadGEM2-ES | 0.78 | NorESM1-M | 0.23 |
| CMCC-CESM | 0.68 | HadGEM2-CC | 1.10 | NorESM1-ME | 0.27 |
| CESM1-WACCM | 0.16 | GFDL-ESM2M | 0.33 | MRI-CGCM3 | 0.05 |
| CESM1-CAM5 | 0.14 | GFDL-ESM2G | 0.59 | MPI-ESM-MR | 0.62 |
| CESM1-BGC | 0.18 | GFDL-CM3 | 0.90 | MPI-ESM-LR | 0.30 |
| CCSM4 | 0.18 | FIO-ESM | 0.84 | MIROC5 | -0.00 |
| BNU-ESM | 1.06 | FGOALS-g2 | 0.94 | MIROC-ESM | 0.38 |
| ACCESS1-3 | 0.58 | CanESM2 | 0.15 | MIROC-ESM-CHEM | 0.42 |
| ACCESS1-0 | 0.25 | CSIRO-Mk3-6-0 | 0.29 | IPSL-CM5B-LR | 1.39 |

**CMIP5 MMM 0.47**

**Figure G1.** Temperature anomaly (2080-2100 mean minus 1989-2009 mean) in the Amundsen Sea (128W-90W;76S-69S) from 33 CMIP5 models in the RCP85 scenario. Continental shelf temperatures (left panel) are averaged over the area where the sea floor is shallower than 1500m, while offshore temperatures (right panel) are averaged over the rest of the domain. The numbers for individual CMIP5 models and the multi-model mean (MMM) indicate the mean ocean warming in the 500-800m layer.



## Appendix H: MISOMIP original calibration below 300 m depth

**Figure H1.** Same as Fig. 7 but the initial calibration is based on averaged melt rates below 300 m depth.

*Author contributions.* L.F. and N.J. designed the experiments and implemented the melt rate parameterizations in Elmer/Ice. F.G-C. implemented the SSA* in Elmer/Ice. N.M. developed the coupling interface. P.M. developed the ice-shelf routines for evolving geometries. L.F. led the writing, and all authors contributed to the writing and discussion of ideas.



*Competing interests.* The authors declare no competing interests.

*Acknowledgements.* This work was funded by the French National Research Agency (ANR) through the TROIS-AS (ANR-15-CE01-0005-01) and the ANR-15-IDEX-02 projects. The simulations were performed using HPC resources from GENCI-CINES (Grant 2018, project A0040106066).





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
