# Peer review of "Assessment of Sub-Shelf Melting Parameterisations Using the Ocean-Ice Sheet Coupled Model NEMO(v3.6)-Elmer/Ice(v8.3)"

_Geoscientific Model Development, 2019_

## Referee Comment (RC1) · Anonymous Referee #1 · 19 Mar 2019

**Comment on 'Assesment of sub-shelf melting parameterisations using the ocean-ice sheet coupled model NEMO-Elmer/Ice'**

**presented on 15th of February 2018 by Favier et al.**

In the manuscript, Favier et al., introduce the coupled model NEMO-Elmer/Ice and use it to assess the perfomance of sub-shelf melt parameterisations. This is done for an idealized setup of an ice-sheet and ocean system and using one coupled model. Although generalisation of the results to different geometries should be made with care, this study is of great interest for ice-sheet modellers relying on such parameterisations and reveals the advantages and disadvantages of the individual parameterisations ranging from simple linear parameterisation to the more complex plume and box model parameterisations. I have a few major comments on the manuscript:

**Major comments:**

- **Figures 5 and 6:**
  It is currently difficult to follow the reasoning of the main results because the figures are hard to read. The different model results are tricky to distinguish and partly not visible at all. My proposition is to split both figures into two figures each, one containing the information from the simple parameterisations and one from the plume and box model parameterisations.
  Some small changes could help to improve the understanding: (a) increasing/decreasing the dash length for the BME parameterisations with increasing number of boxes and for PME with increasing numbers, (b) if possible increased linewidth and legend font size would be great, (c) the $\pm 50\%$ range of ocean model results could be indicated in grey.
  In some cases, data is missing, e.g., in Fig. 5 panel (C) $M+_{700}$ is missing after 70yrs, in panel (E) $M_{lin}$ is hard to see, in panel (F) $BM_{5,500}$ stops after 50 years. In Figure 6, panel (B) $BM_{2,700}$ stops after 60 years, in (D), $M_{lin,700}$ is missing after 80 years and in panel (E), $M_{lin}$ (red) is not visible.

- **Plume parameterization:**

  More detail is needed: the explanation of why the melt rates show this patten of no melting near the grounding line and then increase towards the calving front without a melt-peak and decline afterwards is unclear to me, see also comment on page 17, lines 6-9.

  Given that its effecive grounding line depth is always the central grounding line, I would expect the PME3 parameterisation to yield results along the center line $y = 40$km that are similar to a line plume model. In Figure 8 (g) of Lazeroms et al. 2019 (doi:10.1175/JPO-D-18-0131.1) melt rates calculated with a comparable plume parameterisation and with a full plume model are shown for PIG with a melt rate peak around 15km and a decrease afterwards. Is this pattern different from the pattern you find here because of the higher $T_0$ value used (1 degree at depth versus $-1$ degree)? If yes, how could this be improved?

**Minor comments**

- page 4, line 12: Please check the reference to Schoof 2007, 'Ice sheet grounding line dynamics: Steady states, stability, and hysteresis'.

- page 8, line 19-20: Please clarify which formulation you mean.

- page 11, line 31: It would be great to have here a short explanation what the second parameterisation is about.

- page 12, line 5: Please add $\alpha$ in formula (7) to make it easier to understand its purpose, e.g., in Table 3.

- page 12, line 9: You could refer here additionally to the Appendix where you explain the effective grounding line depth.

- page 12, line 24ff: Please clarify that you explain the calibration of the coupled runs.

- Figure 3: Warm1 profile is missing in Panel (E). Generally, the details of this figure are hard to see. Could you maybe increase linewidth? And make the color schemes more intuitive by , e.g., using blue for the "Cold" scenario?

- page 15, line 20: Maybe add the missing plots in a supplement.

- page 15, line 30: The pattern in the TYP-10m experiment looks different from the other coupled runs as it shows melting at the opposite margin of the ice-shelf - why could this be the case? And what causes the wave-like pattern in the basal melt rates of the coupled model?

- page 15, line 33: 'occurs' instead of 'falls'?

- Figure 4, Appendix D4: It's not clear why there is no melting in the area $y \leq 40$m for PME4: the algorithm (as described in the Appendix) would identify the closest grounding line point as the effective grounding line depth for points in this region. An example is shown in Figure E1 (C, example 2). I guess that those points are excluded based on the criteria for PME1?

- page 16, line 3: 'are similar by construction'.

- page 17, lines 6-9: Please clarify: I do not understand why a plume rising from only a limited number of directions reduces the melt rates, since, as explained in the Appendix and in Lazeroms et al., 2018, the effective grounding line depth is calculated as an average and similarly the effective slope is an average value (or the local gradient)?
  Also, I would expect the central grounding line point to be generally part of a 'valid' direction, since it is the deepest point of the ice shelf - how can then the melting at the 'inner sides' of the ice shelf increase, because the plumes can emerge from 'more deeper portions'?
  And third, it is not clear to me how a 'combination' of more plumes can generate higher melting towards the ice front? Shouldn't in this case, because plumes can emerge also from shallower grounding line regions, the effective (average) grounding line depth be shallower than close to the grounding line? Wouldn't in this case the thermal driving be also lower (WARM profile)? Then the higher melting must relate to the plume scaling and the dimensionless melt rate curve $\hat{M}(\hat{X})$ or $g(\alpha)$?

- page 17, line 25: I think 'latter' and 'former' are switched.

- Figure 5: What causes the variations in the coupled model run in basal melt fluxes in comparison to the parameterisations?

- Figure 5 B: Some parameterisations show a decrease after $\approx 70$ years. Is this because ice-shelf area is lost?

- page 20, line 18: $M_{quad,700}$ does not seem to do well for Warm3.

- page 20, line 26-27: '.. reflects the increase in thermal forcing compared to the depth-dependent forcing.' I do not understand your statement here: the thermal forcing for 500m depth is lower at depth and higer towards the surface and seems on average to be comparable to the thermal forcing in the depth-dependent parameterisation (Figure 3)?

- page 20, line 27-29: 'However, if the given...'. Please clarify: how does this statement relate to the result that in the 'Cold0', 'Warm3' experiments, the SLC for 500m is higher for all parameterisations while in the 'Warm0,1,2' all parameterisations using 700m have higher SLC?

- page 21, line 24, Figure 7: Please indicate that the RMSE is calculated by summing the deviations of SLC over all experiments (if this is true).

- Figure E1: If one doesn't know that the difference between PME1 and PME2 is how the calculation of the effective angle, it's confusing that Panel (A) shows both parameterisations.

- Appendix A: Please explain $u_m^b$ and $u_b$.

- page 27, line 13: Please define $\theta$.

- page 29, line 3: I think with 'checkerboard noise' you refer to Fig. 4 ?

- page 30, line 11: Since the formulas are not complicated, it would be helpful to add them here.

**Technical issues**

- page 1, line 20: 'ice mass loss' and 'ice-shelf thinning' are exchanged?

- page 2, line 3: 'lowering of grounded ice surface'?

- page 4, line 4: 'controlled by Glen's flow law'.

- page 4, line 9: switch 'Seroussi and Morlighem, 2018' and 'equivalent to the SEP3 method in'.

- page 7, line 23: 'this' too much.

- page 11, line 17: ice-shelf basal slope $\theta$.

- page 11, line 25: Appendix C.

- page 12, line 25: Figure 3.

- Figure 3: Panels (C), (D), (E) are switched to (D),(E),(F).

- page 31, line 4: anti-clockwise

- Figure E1: 'used in the present paper'.

- page 23, line 34: 'Ekman pumping'.

- page 24, line 33: 'multiple'.

- page 27, Appendix B should be Appendix C.

- page 31, Figure E1 should be D1.

- page 32, Sections E1 and E2 should be D5, D6.

- page 33, Appendix F should be E.

- page 33, Figure G1 should be F1 and 'Nico-' can be deleted in the title.

- page 34, Appendix G

- page 36, line 20: 'received' too much.

---

## Referee Comment (RC2) · Anonymous Referee #2 · 22 Mar 2019

The authors present the first results from a newly developed offline coupling between the ice model Elmer/Ice and ocean model NEMO. They use the standardized MIS-MIP/MISOMIP intercomparison framework to assess the impact of ocean melting on ice dynamics for a range of (idealized) future ocean conditions, demonstrating the capabilities of their model. They then use these (sophisticated) coupled ice-ocean results as a benchmark to assess the performance of a range of (simplified) ice-shelf melting parameterization that are commonly used in a stand-alone ice models.

This is a solid piece of work, which recognises the importance and complexities of simulating ice-ocean interactions in Antarctic ice shelf cavities, and the need for careful model development and validation. The manuscript presents the first comparison between results from a coupled ice-ocean model and a comprehensive set of commonly used meltrate parameterizations, making this a timely and valuable reference for further research. The model development sections and appendices contain sufficient amounts of (technical) detail, the methods are sound and in line with previously published developments, and the experiments are explained in a comprehensive way. The authors demonstrate the capabilities of their new coupled ice-ocean model, and pave the way for further research. I highly recommend this work for publication in GMD, although I would like the suggest a few points for further clarification/improvement.

1) For the calibration of the melt calculations you use the WARM ocean conditions, but adopt different criteria to fix the exchange coefficients in the coupled (<melt>=30m/yr below 300m) and parameterized (<melt>=8.5mr/yr over the entire ice shelf) melt calculations. This -somewhat ad hoc- choice of calibration subsequently has large effects on the results (comparing Figure 7 and H1) and it therefore seems rather important. Could you please clarify why you do not adopt a universal calibration for all (parameterized and coupled) methods, e.g. <melt>=8.5mr/yr over the entire ice shelf and what would inform such a calibration? For a universal calibration, the initial differences in SLC between different melt calculations are fully attributed to differences in the spatial distribution of melt, at least in the WARM scenario, rather than spurious effects due to the calibration method.

2) On a similar note, I wasn't expecting the spread in total melting between different melt parameterizations (Figure 5) to remain more-or-less constant through time. I was expecting a divergence, with most parameterizations doing progressively 'worse' over time, compared to coupled simulations. Do you have any insights in to why that is?

I understand that the initial spread is defined by how sensitive the melt parameterizations are to the forcing for a given geometry, with the constraint that total melting is the same for all parameterization in the WARM (calibration) scenario. To disentangle the geometrical feedbacks and the initial sensitivity to ocean forcing, I think it would be

instructive to see a plot similar to the panels in Figure 5, but for the WARM scenario. By nature of your calibration, all simulations should agree on the total melting at time 0, and (perhaps) divert for t>0 due to ice-ocean feedbacks. By imposing a common starting point, you will be able to unambiguously identify which parameterizations are 'close to' or 'far away' from the coupled simulations for that particular forcing.

3) About the RMSD criterion (Figure 7) that you use to assess the performance of parameterized melt rates compared to coupled ice-ocean simulations, I wonder if this criterion might be too simple and perhaps even misleading. As you explain, the spatial distribution of melt rates for a given geometry will, to a large degree, control the dynamic response of the glacier. However, this is not clearly captured by the RMSD criterion. In particular, you could identify parameterizations with a low RMSD as 'performing well', but the spatial distribution of melt could be totally wrong, and therfore the RMSD is low for the wrong reasons. As a result, this parameterization might not be suitable for other (more complex) geometries. Perhaps a simple RMSD criterion should be supplemented by a measure of how well the spatial melt distribution compares to the coupled scenario, making the assessment more robust. To achieve this, it would be instructive to see Figure 4 but after 50 years of simulations.

Finally, here is a list of some smaller comments, typos etc.

p1, l15 shelt → shelf

p4, l19 'floating nodes only': please clarify if you impose melt for nodes in partially grounded elements

p5, l1 different ocean models will use different ways to parameterize a 'boundary layer' and calculate u_TLB. Perhaps you could be more specific here about u_TLB, unless this methodology has been published elsewhere?

p5, l17 do you always average over the entire coupling period, or do you use the final week/month/... of that period?

p5, l22 'too thin to be captured by NEMO': could you be more specific please? Do you impose a minimum water column thickness? Do you adjust your geometry to allow for this etc?

p6, l10-12 You say you have shown convergence with coupling timestep, but in Figure A1 all results fall on top of each other. To fully show convergence, you need to present results from eg 48 and 24 months, and show that they 'converge' to the solution for 12, ... months. On a similar note, you present results for 1 particular scenario. In the caption of Figure A1 you say this is the COM-Ocean1 experiment, but I'm not sure if you mean Ocean1r or Ocean1ra? As the total melt goes down over time, I'm assuming this is a 1ra scenario with cold forcing? This could be important, as convergence might be harder to achieve in a warm ocean scenario?

p6, l17 As the calibration procedure is so important, perhaps you could provide a 1-sentence summary of the protocol from (Asay-Davis et al., 2016)?

p6, Table1 Of course this list of experiments is not/cannot be exhaustive, but it would be nice to have some more motivation for the choice of these specific experiments. Do they somehow represent end-members that allow you to put generous 'error bars' on the coupled results? For example, why are you interested in constant tidal velocities, knowing that Pine Island is not subjected to strong tidal currents? What defines T_CPL and why is it different for the different scenarios, given that you claim 'convergence' below 12 months. Why do you not consider changes in the drag coefficient, etc?

p8, l10 'inspired BY Jourdain et al. (2017) WHO ...

p10, l9 'to the local pressure' → 'ON the local pressure'

p12, l25 do you mean figure 2?

p14 Perhaps you can point out that you do not impose any 'density compensation' by changing the salinity for each temperature profile, so the density profile will be different in each scenario. Figure 3. The labels for the different sub figures got mixed up in the

caption. I guess D should be C, E should be D and F should be E?

Figure 4. In the top row, one panel shows contours. Are they ice draft?

p17, l9 conbined → combined

p17, l20-25: How do you explain this difference in timescale between the initial melting pulse in the parameterized vs coupled case?

p17, l25: Do you mean 'lasts longer for the former, about 20a, than for the latter, about 5a'?

p17, l26: 'a melting minimum': This is not entirely obvious to me, it is certainly not 'universal', e.g. PME1 and BM5_500 seem to be monotonically decreasing?

p17, l30: surface → sea surface

Figure 5. Caption: 'Same as figure 6': please describe here, and refer in the caption of Figure 6 to Figure 5. The solid light grey curves are very hard to see, so I would consider using a darker shade or a different colour. Also, I would like to see these curves included in the figure legend. I'm also finding it impossible to distinguish between Mlin, PME1 and PME4 as they all print in pink. The same happens for Mquad and PME2. Perhaps you could consider expanding your range of colours, at the risk of making this figure even harder to interpret?

p20, l5-7 I'm a bit lost here, could you please reformulate this statement?

p21, l5-10: do results converge with a larger number of boxes, or is there an optimal choice for the number of boxes?

p21, l22 and Figure 7. It's unclear to me why you chose 50 years as a time for your performance indicator. As you have 100 years of simulations, why not use 100 years? Also, is there any way you can account for the difference in initial pulse, which lasts 20 years in the parameterized cases compared to 5 years in the coupled setup? This difference might severely bias your performance indicator, altough it is probably a constant offset.

p26, 27: Appendix B is used twice, please check the numbering

p31, l3: remove 'the'

p31, l4: anto → anti and remove 'until'

p33, Appendix F: I'm not sure where this appendix has been refered to in the main text

---

## Author Comment (AC1) · 26 Apr 2019

**Response to anonymous referee RC1 on "Assessment of Sub-Shelf Melting Parameterisations Using the Ocean-Ice Sheet Coupled Model NEMO(v3.6)-Elmer/Ice(v8.3)" by Favier et al.**

April 26, 2019

In the manuscript, Favier et al., introduce the coupled model NEMO-Elmer/Ice and use it to assess the perfomance of sub-shelf melt parameterisations. This is done for an idealized setup of an ice-sheet and ocean system and using one coupled model. Although generalisation of the results to different geometries should be made with care, this study is of great interest for ice-sheet modellers relying on such parameterisations and reveals the advantages and disadvantages of the individual parameterisations ranging from simple linear parameterisation to the more complex plume and box model parameterisations.
We thank the reviewer for this positive comment

I have a few major comments on the manuscript:

**Major comments:**

• Figures 5 and 6: It is currently difficult to follow the reasoning of the main results because the figures are hard to read. The different model results are tricky to distinguish and partly not visible at all. My proposition is to split both figures into two figures each, one containing the information from the simple parameterisations and one from the plume and box model parameterisations. Some small changes could help to improve the understanding: (a) increasing/decreasing the dash length for the BME parameterisations with increasing number of boxes and for PME with increasing numbers, (b) if possible increased linewidth and legend font size would be great, (c) the ±50% range of ocean model results could be indicated in grey. In some cases, data is missing, e.g., in Fig. 5 panel (C) M+700 is missing after 70yrs, in panel (E) Mlin is hard to see, in panel (F) BM5,500 stops after 50 years. In Figure 6, panel (B) BM2,700 stops after 60 years, in (D), Mlin,700 is missing after 80 years and in panel (E), Mlin (red) is not visible.
We agree with the reviewer and have split Fig 5 and Fig 6 accordingly. We have also followed the other recommendations and added the missing results.

• Plume parameterization: More detail is needed: the explanation of why the melt rates show this patten of no melting near the grounding line and then increase towards the calving front without a melt-peak and decline afterwards is unclear to me, see also comment on page 17, lines 6-9. Given that its effecive grounding line depth is always the central grounding line, I would expect the PME3 parameterisation to yield results along the center line y = 40km that are similar to a line plume model. In Figure 8 (g) of Lazeroms et al. 2019 (doi:10.1175/JPOD-18-0131.1) melt rates calculated with a comparable plume parameterisation and with a full plume model are shown for PIG with a melt rate peak around 15km and a decrease afterwards. Is this pattern different from the pattern you find here because of the higher T0 value used (1 degree at depth versus -1 degree)? If yes, how could this be improved?
This is a very interesting remark as it highlights the effect of the different choices in the implementation of the plume parameterisation. The PME3 accounts for the slope between the grounding line and the draft point as the effective angle, while the approach of PME1 (that is developed in Lazeroms et al, 2018, and also in the Lazeroms et al, 2019) accounts for the local slope. The two curves are thus difficult to compare, and also because they are the result of different environmental conditions of geometry, ambient temperature and salinity. In that sense, Fig. 8g of the Lazeroms et al, 2019, may not be the best to be compared with. Here below, you will see Fig. 1, which represents PME3 and PME5 (Fig C1 and Ap. D of the paper). The difference between PME3 and PME5 is in the definition of the effective slope, the local gradient being considered in PME5 instead of the slope between the grounding line and the draft point. We have chosen to represent the $Cold_0$ in Fig. 1 because the temperature of -1.5 is closer to -1.9 used in Lazeroms et al, 2019 (maybe you can consider their Fig. 5d to be compared with). To keep the paper clear, we have decided not to give those technical details

[Figure]

Figure 1: Melt rates obtained at $t = 0$ by the PME3 and PME5 parameterisations forced by the $Cold_0$ scenario. PME3 uses the slope between the draft point and the grounding line to calculate the effective angle, while PME5 uses the local gradient to do so. Ice shelf geometry in the top raw, PME3 in the middle raw and PME5 in the bottom raw.

in the paper. And, to conclude, we do have a melting peak in our simulations but the melt rates distribution will depend on the way the effective slope is accounted for, the temperatures, the thickness at the calving front, among other things, which for us opens the road to further investigations.

**Minor comments**

• page 4, line 12: Please check the reference to Schoof 2007, 'Ice sheet grounding line dynamics: Steady states, stability, and hysteresis'.
We have corrected the reference

• page 8, line 19-20: Please clarify which formulation you mean.
We mean when $T_o$ and $S_o$ are depth dependent, which we have clarified in the text.

• page 11, line 31: It would be great to have here a short explanation what the second parameterisation is about.
To clarify the text, we now only mention that the methods related to the plume parameterisation are all based on different calculations of effective values for the grounding line depth and the basal slope. All the specific details to each method are now in Ap. D only.

• page 12, line 5: Please add $\alpha$ in formula (7) to make it easier to understand its purpose, e.g., in Table 3.
Done

• page 12, line 9: You could refer here additionally to the Appendix where you explain the effective grounding line depth.

Done

• page 12, line 24: Please clarify that you explain the calibration of the coupled runs.
The initial melt from which start parameterised and coupled simulations is actually defined with an ocean spin-up (like depicted in Fig1) that is also the starting point of coupled simulations. We have rephrased this part to make it clearer

• Figure 3: Warm1 profile is missing in Panel (E). Generally, the details of this figure are hard to see. Could you maybe increase linewidth? And make the color schemes more intuitive by , e.g., using blue for the "Cold" scenario?
The Warm1 and Warm2 profiles are actually equal in panel E. We now mention it in the caption. For the rest, we have followed the suggestions of the reviewer.

• page 15, line 20: Maybe add the missing plots in a supplement.
We already have a lot of figures so we prefer not. Moreover, the parameterised melt that we don't show here are relatively easy to plot, as opposed to the more complicated pattern that we already show in Figure 4.

• page 15, line 30: The pattern in the TYP-10m experiment looks different from the other coupled runs as it shows melting at the opposite margin of the ice-shelf - why could this be the case?
It is not clear why the melting pattern of "TYP-10m" is different. The pattern suggests that more melting occurs when the warm water mass enters the cavity in TYP-10m (lower part of the plots), so that no more heat is available to significantly melt the ice along the outflowing jet (upper part of the plots). TYP-10m has a thinner TBL but stronger $\Gamma_T$ coefficient. It is possible that the higher $\Gamma_T$ coefficient make the exchange more efficient when the water masses enter the cavity, but the thinner TBL limits the amount of ocean heat immediately available for ice melting, so no more heat is available along the outflowing jet. Preliminary results of the ISOMIP+ intercomparison (Asay-Davis et al. 2016) indicate that both patterns are found across the existing ocean models. We have not added this discussion to keep the manuscript relatively short and with a clear focus.

And what causes the wave-like pattern in the basal melt rates of the coupled model?
The wave pattern is commonly found in the z-coordinate models taking part to ISOMIP+ (preliminary results). This is possibly related to the partial steps used for the upper levels, which makes that TBL averages can average one or several levels depending on the thickness of partial cells. This is an aspect of z-coordinate models that will need to be investigated.

• page 15, line 33: 'occurs' instead of 'falls'?
Changed

• Figure 4, Appendix D4: It's not clear why there is no melting in the area $y < 40m$ for PME4: the algorithm (as described in the Appendix) would identify the closest grounding line point as the effective grounding line depth for points in this region. An example is shown in Figure E1 (C, example 2). I guess that those points are excluded based on the criteria for PME1?
For $y < 40m$, the melting are either 0 or low because either the unique direction found by the algorithm leads to a higher grounding line than the draft point, or the slope is low, respectively

• page 16, line 3: 'are similar by construction'.
Changed

• page 17, lines 6-9: Please clarify: I do not understand why a plume rising from only a limited number of directions reduces the melt rates, since, as explained in the Appendix and in Lazeroms et al., 2018, the effective grounding line depth is calculated as an average and similarly the effective slope is an average value (or the local gradient)? Also, I would expect the central grounding line point to be generally part of a 'valid' direction, since it is the deepest point of the ice shelf - how can then the melting at the 'inner sides' of the ice shelf increase, because the plumes can emerge from 'more deeper portions'? And third, it is not clear to me how a 'combination' of more plumes can generate higher melting towards the ice front? Shouldn't in this case, because plumes can emerge also from shallower grounding line regions, the effective (average) grounding line depth be shallower than close to the grounding line? Wouldn't in this case the thermal driving be also lower (WARM profile)? Then the higher melting must relate to the plume scaling and the dimensionless melt rate curve $M(X)$ or $g(\alpha)$?
You are right, the melting quantity doesn't depend directly on the number of directions. We have changed the sentences accordingly

• page 17, line 25: I think 'latter' and 'former' are switched.
Right, changed

• Figure 5: What causes the variations in the coupled model run in basal melt fluxes in comparison to the parameterisations?
These variations can also be seen in Figure A1. Given the absence of atmospheric and sea ice forcing, these variations must be related to an internal mode associated with the geometry of the closed ocean domain. For exemple, the 2-3 year period could be the typical advection time around the domain. In any case, this is very specific to the MISOMIP geometry and has not been investigated further.

• Figure 5 B: Some parameterisations show a decrease after ≈ 70 years. Is this because ice-shelf area is lost?
Yes, this is one of the reasons. This is why we set up the performance indicator at 50 years. We already mentioned this point in the first paragraph of the discussion

• page 20, line 18: Mquad,700 does not seem to do well for Warm3.
Right, modified

• page 20, line 26-27: '.. reflects the increase in thermal forcing compared to the depth dependent forcing.' I do not understand your statement here: the thermal forcing for 500m depth is lower at depth and higer towards the surface and seems on average to be comparable to the thermal forcing in the depth-dependent parameterisation (Figure 3)?
We have rephrased. The main idea here is that more thermal forcing does not mean more SLC, because the initial calibration step will result in a lower multiplicative coefficient ($\gamma_T$ or $\alpha$ for the $PME_i$ parameterisations).

• page 20, line 27-29: 'However, if the given...'. Please clarify: how does this statement relate to the result that in the 'Cold0', 'Warm3' experiments, the SLC for 500m is higher for all parameterisations while in the 'Warm0,1,2' all parameterisations using 700m have higher SLC?
We have removed this sentence

• page 21, line 24, Figure 7: Please indicate that the RMSE is calculated by summing the deviations of SLC over all experiments (if this is true).
The way it is calculated is written in the previous line in the main text

• Figure E1: If one doesn't know that the difference between PME1 and PME2 is how the calculation of the effective angle, it's confusing that Panel (A) shows both parameterisations.
Yes, we have added a sentence in the caption to recall that difference

• Appendix A: Please explain ub m and ub.
This was a typo. This is u_b in both case, but one is to the power of m

• page 27, line 13: Please define $\theta$
Done here and also in the main text

• page 29, line 3: I think with 'checkerboard noise' you refer to Fig. 4 ?
Right, modified

• page 30, line 11: Since the formulas are not complicated, it would be helpful to add them here.
We have done as suggested.

**Technical issues**

• page 1, line 20: 'ice mass loss' and 'ice-shelf thinning' are exchanged?
Indeed, modified
• page 2, line 3: 'lowering of grounded ice surface'?
Modified
• page 4, line 4: 'controlled by Glen's flow law'.
Changed
• page 4, line 9: switch 'Seroussi and Morlighem, 2018' and 'equivalent to the SEP3 method in'.
Right, modified

• page 7, line 23: 'this' too much.
Modified
• page 11, line 17: ice-shelf basal slope $\theta$
Done
• page 11, line 25: Appendix C.
Done
• page 12, line 25: Figure 3.
Changed
• Figure 3: Panels (C), (D), (E) are switched to (D),(E),(F).
Done
• page 31, line 4: anti-clockwise
Done
• Figure E1: 'used in the present paper'.
Done
• page 23, line 34: 'Ekman pumping'.
Done
• page 24, line 33: 'multiple'.
Done
• page 27, Appendix B should be Appendix C.
Done
• page 31, Figure E1 should be D1.
Done
• page 32, Sections E1 and E2 should be D5, D6.
Done
• page 33, Appendix F should be E.
Done
• page 33, Figure G1 should be F1 and 'Nico-' can be deleted in the title.
Done.
• page 34, Appendix G
Done.
• page 36, line 20: 'received' too much.
Removed

---

## Author Comment (AC2) · 26 Apr 2019

**Response to anonymous referee RC2 on "Assessment of Sub-Shelf Melting Parameterisations Using the Ocean-Ice Sheet Coupled Model NEMO(v3.6)-Elmer/Ice(v8.3)" by Favier et al.**

April 26, 2019

The authors present the first results from a newly developed offline coupling between the ice model Elmer/Ice and ocean model NEMO. They use the standardized MISMIP/MISOMIP intercomparison framework to assess the impact of ocean melting on ice dynamics for a range of (idealized) future ocean conditions, demonstrating the capabilities of their model. They then use these (sophisticated) coupled ice-ocean results as a benchmark to assess the performance of a range of (simplified) ice-shelf melting parameterization that are commonly used in a stand-alone ice models. This is a solid piece of work, which recognises the importance and complexities of simulating ice-ocean interactions in Antarctic ice shelf cavities, and the need for careful model development and validation. The manuscript presents the first comparison between results from a coupled ice-ocean model and a comprehensive set of commonly used meltrate parameterizations, making this a timely and valuable reference for further research. The model development sections and appendices contain sufficient amounts of (technical) detail, the methods are sound and in line with previously published developments, and the experiments are explained in a comprehensive way. The authors demonstrate the capabilities of their new coupled ice-ocean model, and pave the way for further research. I highly recommend this work for publication in GMD, although I would like the suggest a few points for further clarification/improvement.
We are very happy with this positive comment

1) For the calibration of the melt calculations you use the WARM ocean conditions, but adopt different criteria to fix the exchange coefficients in the coupled (¡melt¿=30m/yr below 300m) and parameterized (¡melt¿=8.5mr/yr over the entire ice shelf) melt calculations. This -somewhat ad hoc- choice of calibration subsequently has large effects on the results (comparing Figure 7 and H1) and it therefore seems rather important. Could you please clarify why you do not adopt a universal calibration for all (parameterized and coupled) methods, e.g. ¡melt¿=8.5mr/yr over the entire ice shelf and what would inform such a calibration? For a universal calibration, the initial differences in SLC between different melt calculations are fully attributed to differences in the spatial distribution of melt, at least in the WARM scenario, rather than spurious effects due to the calibration method.
The target of 30 m/a of melting below 300 m depth using the WARM ocean conditions is part of the MISOMIP protocol. We simply used this protocol because the WARM ocean conditions lead to a quick relaxation of the ocean, which allows to find the calibration without cumbersome calculations. However, when forced by the WARM ocean conditions, all the sub-shelf melt parameterisations produce substantial melting above 300 m depth, as opposed to ocean models that produce small melt rates. We thus chose to take the average of melting without depth limitation, but in the end, we have the same melting average of 8.5 m/a (+/- 1) for all parameterised and coupled simulations. All this was explained, and has now been clarified in response to reviewer 1, in Section 3.2.

2) On a similar note, I wasn't expecting the spread in total melting between different melt parameterizations (Figure 5) to remain more-or-less constant through time. I was expecting a divergence, with most parameterizations doing progressively 'worse' over time, compared to coupled simulations. Do you have any insights in to why that is?
We are not sure about what the reviewer means here. In terms of sea-level contribution, it is quite clear from revised Fig.7-8, that the distance between the parameterizations and the coupled model and the distance between different parameterizations increase in time. In terms of melt rates it is different because melt rates remain highly constrained by the temperature scenarios.

I understand that the initial spread is defined by how sensitive the melt parameterizations are to the forcing for a given geometry, with the constraint that total melting is the same for all parameterization in the WARM (calibration) scenario. To disentangle the geometrical feedbacks and the initial sensitivity to ocean forcing, I

think it would be instructive to see a plot similar to the panels in Figure 5, but for the WARM scenario. By nature of your calibration, all simulations should agree on the total melting at time 0, and (perhaps) divert for t¿0 due to ice-ocean feedbacks. By imposing a common starting point, you will be able to unambiguously identify which parameterizations are 'close to' or 'far away' from the coupled simulations for that particular forcing.

This is an interesting suggestion. Actually, there are two reasons why melting will divert at t¿0: (1) the ice/ocean feedback, i.e. how the change in ice draft depth modifies melt rates, and (2) the various melt parameterizations all have different sensitivities to warming, and as restoring temperature evolves, melt rates will diverge even without ice/ocean feedback. So investigating this would also probably require simulations with fixed cavities. This is interesting but our approach is to evaluate melting through the ice sheet response because we don't want to decide a priori where melt pattern has to be accurate. Overall, this suggestion seems to be quite disconnected to our approach, and to keep the paper relatively short and clear, we have decided not to follow the reviewer's suggestion.

3) About the RMSD criterion (Figure 7) that you use to assess the performance of parameterized melt rates compared to coupled ice-ocean simulations, I wonder if this criterion might be too simple and perhaps even misleading. As you explain, the spatial distribution of melt rates for a given geometry will, to a large degree, control the dynamic response of the glacier. However, this is not clearly captured by the RMSD criterion. In particular, you could identify parameterizations with a low RMSD as 'performing well', but the spatial distribution of melt could be totally wrong, and therfore the RMSD is low for the wrong reasons. As a result, this parameterization might not be suitable for other (more complex) geometries. Perhaps a simple RMSD criterion should be supplemented by a measure of how well the spatial melt distribution compares to the coupled scenario, making the assessment more robust. To achieve this, it would be instructive to see Figure 4 but after 50 years of simulations.

This is clear that the distribution of melting from parameterised simulations and coupled simulations are quite different, which is actually discussed in the paper ("All parameterisations yield too large melt rates in thin ice areas and too small melt rates ..."). However, we do not agree that the RMSD is misleading even though it is not ideal. Assessing the spatial distribution of melt would also be not ideal because it is difficult to identify which part of the melting pattern is important for the ice dynamics (e.g. see "tele-butressing" in Reese et al. 2018b). For this reason, we have decided to evaluate the parameterizations through an ice-sheet simulation rather than through the melting characteristics.

Finally, here is a list of some smaller comments, typos etc.
p1, l15 shelt → shelf
Changed

p4, l19 'floating nodes only': please clarify if you impose melt for nodes in partially grounded elements
Even though the melting is 0 at the last grounded point and 1 at the first floating point, the finite element method will give an averaged solution for the element containing these two (or three here) points. Thus the last grounded node may be affected by melting if the thickness is sufficiently modified to make it float. Otherwise, the last grounded node will stay grounded. As the treatment of melting is the same by Elmer/Ice in both parameterised and coupled simulation, we have not added those technical points but now simply say "Melting is applied to floating nodes but not to grounded nodes, meaning that the first floating element (partially or not) may be affected by melting."

p5, l1 different ocean models will use different ways to parameterize a 'boundary layer' and calculate $u_{TBL}$. Perhaps you could be more specific here about $u_{TBL}$, unless this methodology has been published elsewhere?
As described by Mathiot et al. (2017), $u_{TBL}$ is averaged over a constant thickness, assumed to represent the thickness of the TBL. We added the reference in the text.

p5, l17 do you always average over the entire coupling period, or do you use the final week/month/... of that period?
Melt rates are always averaged over the entire coupling period in order to conserve mass as much as possible, which we added in the paper. By contrast, the final ice draft of Elmer/Ice's coupling period is always sent to NEMO.

p5, l22 'too thin to be captured by NEMO': could you be more specific please? Do you impose a minimum water column thickness? Do you adjust your geometry to allow for this etc?
We impose a minimum water column thickness of 20m, which allows NEMO to have a minimum of two vertical cells under the partial-cells condition, which we added to the paper. If the new water column opened by Elmer/Ice is thinner than 20m, it is not simulated by NEMO, i.e. zero melt rate is sent back to Elmer/Ice.

p6, l10-12 You say you have shown convergence with coupling timestep, but in Figure A1 all results fall on top of each other. To fully show convergence, you need to present results from eg 48 and 24 months, and show that they 'converge' to the solution for 12, . . . months. On a similar note, you present results for 1 particular scenario. In the caption of Figure A1 you say this is the COM-Ocean1 experiment, but I'm not sure if you mean Ocean1r or Ocean1ra? As the total melt goes down over time, I'm assuming this is a 1ra scenario with cold forcing? This could be important, as convergence might be harder to achieve in a warm ocean scenario?
This is the retreat IceOcean1r scenario (warm), which has been specified. The initial strong melt rate is related to the initialization method in MISOMIP, but after 5years, melt increases over 95 years. We do not see the point of showing results for 24-month and 48-month coupling periods because we never use such coupling period. Of course, if we make the coupling period very long, it will make a difference, but the point is more that taking any coupling period between 1-month and 12-month does not affect the result. Also, nowhere in the text we wrote that we have convergence of the solution when decreasing the coupling period. We simply mentioned that there is no sensitivity to the coupling period in the 1-12 months range.

p6, l17 As the calibration procedure is so important, perhaps you could provide a 1sentence summary of the protocol from (Asay-Davis et al., 2016)?
We have added the sentence "more details of the protocol relevant to our study are given in Sec. 2.2.3, and the protocol is fully described in Asay-Davis, 2016, Sec. 3.2.1" at the end of Section 2.2.3, and in Section 3.2.1 we have added the sentence "The remaining steps of our calibration, described here below, differ from the ISOMIP+ protocol and are specific to our study" so it is clear what is part of the ISOMIP+ protocol and what is specific to our calibration. The relevant details of the protocol are still described at the previous sentence.

p6, Table1 Of course this list of experiments is not/cannot be exhaustive, but it would be nice to have some more motivation for the choice of these specific experiments. Do they somehow represent end-members that allow you to put generous 'error bars' on the coupled results? For example, why are you interested in constant tidal velocities, knowing that Pine Island is not subjected to strong tidal currents? What defines $T_{CPL}$ and why is it different for the different scenarios, given that you claim 'convergence' below 12 months. Why do you not consider changes in the drag coefficient, etc?
The coupling frequency is different but has no effect. We do not change the drag coefficient $C_d$ because this is somewhat equivalent to changing $\Gamma_T$ which is our tuning coefficient (see Jourdain et al. 2017). The rest is based on our experience of what is likely to affect melt rates: vertical mixing, vertical resolution, and horizontal resolution. Tidal velocities directly affect heat exchange and were therefore thought to be important. We agree that tides are unimportant for Pine Island (e.g. Jourdain et al. 2018), but this idealized cavity represents a range of small cavities in warm and cold environments, not only Pine Island, although it was initially chosen to mimic Pine Island in the MISOMIP protocol.

p8, l10 'inspired BY Jourdain et al. (2017) WHO ...
Changed

p10, l9 'to the local pressure' → 'ON the local pressure'
Changed

p12, l25 do you mean figure 2?
Yes, changed

p14 Perhaps you can point out that you do not impose any 'density compensation' by changing the salinity for each temperature profile, so the density profile will be different in each scenario.
We have added the following sentence where the scenrions are described: "Note that none of the temperature profiles account for a salinity compensation (as opposed to the MISOMIP protocol), so the density profile is different in each scenario"

Figure 3. The labels for the different sub figures got mixed up in the caption. I guess D should be C, E should be D and F should be E?
Yes, corrected

Figure 4. In the top row, one panel shows contours. Are they ice draft?
Yes, it is now clearly specified

p17, l9 conbined → combined
We have changed this part as a response to the other reviewer

p17, l20-25: How do you explain this difference in timescale between the initial melting pulse in the parameterized vs coupled case?

We think that when a melt pulse in the coupled model, a lot of fresh water is added to the system that will further decrease melting. Such feedback is either not or poorly accounted for in the parameterizations. We have added this explanation to the paper.

p17, l25: Do you mean 'lasts longer for the former, about 20a, than for the latter, about 5a'?

Yes, this typo was also pointed out by the other reviewer. We have corrected it

p17, l26: 'a melting minimum': This is not entirely obvious to me, it is certainly not 'universal', e.g. PME1 and $BM5_{500}$ seem to be monotonically decreasing?

This is true, we have rephrased the paragraph

p17, l30: surface → sea surface

Changed

Figure 5. Caption: 'Same as figure 6': please describe here, and refer in the caption of Figure 6 to Figure 5. The solid light grey curves are very hard to see, so I would consider using a darker shade or a different colour. Also, I would like to see these curves included in the figure legend. I'm also finding it impossible to distinguish between Mlin, PME1 and PME4 as they all print in pink. The same happens for Mquad and PME2. Perhaps you could consider expanding your range of colours, at the risk of making this figure even harder to interpret?

This was also a point from the other reviewer. For each of these figures, we have now two new figures for simple and more complex parameterisations. It should be easier to interpret and understand now.

p20, l5-7 I'm a bit lost here, could you please reformulate this statement?

We have removed this part

p21, l5-10: do results converge with a larger number of boxes, or is there an optimal choice for the number of boxes?

In the SI of Reese et al., 2018, a convergence is showed above 5 boxes. In our study we don't find a convergence, but it is unclear whether this is related to different resolutions or to the specificities of the MISOMIP geometry.

p21, l22 and Figure 7. It's unclear to me why you chose 50 years as a time for your performance indicator. As you have 100 years of simulations, why not use 100 years?

As we say in the first paragraph of the discussion, a significant part of the ice shelf is melted out by the parameterisations after 50 years of simulations, which makes comparisons difficult after this period since this is less the case for coupled simulations

Also, is there any way you can account for the difference in initial pulse, which lasts 20 years in the parameterized cases compared to 5 years in the coupled setup? This difference might severely bias your performance indicator, although it is probably a constant offset.

The duration of this initial pulse is not equal between the different experiments, especially for parameterised simulations, thus we would not know what period to use. We nonetheless agree that assessing the ability of a parameterisation to cope with such a pulse would be interesting.

p26, 27: Appendix B is used twice, please check the numbering

Checked and corrected

p31, l3: remove 'the'

Done

p31, l4: anto → anti and remove 'until'

Done

p33, Appendix F: I'm not sure where this appendix has been refered to in the main text

The figure E1 was cited instead, we have modified it and now cite Ap E

---

## Author Response (AR2)

**Response to Reviewer 1**

May 21, 2019

page 1, line 17-18: This is formulated ambiguously: do you mean with ".. and apply them to more realistic regional configurations" that further work should try to assess their validity in more realistic setups or that more work is in general needed before one can test them in realistic configurations at all? In the latter case, please specify why.

We replaced the last sentence of the abstract by "Further work is therefore needed to assess the validity of these melting parameterizations in more realistic setups"

page 17, line 33-34: Do you mean that the pulse in the "Cold0" scenario only shows for the coupled runs and not for the parameterisations? Please clarify.

This is correct, we have clarified

page 22, line 2-4: It's hard to tell, but for some coupled runs, melt rates in the "Warm0" case seem to increase first and then drop after 75 years.

Yes indeed. It also seems to be the case for some coupled run of the "Warm1". It's hard to tell why actually.

page 22, line 7-8: "50 to 100 Gt/a"? (about 50Gt/a for PME2 in Cold1, 100Gt/a for coupled in Cold1)

Yes thanks, corrected

page 22, line 8: maybe reformulate to "4 to 12mm of sea-level equivalent mass" since this is an idealized setup

This is right. Changed

page 23, line 5 and following: You could add that both, the plume emulator and the box parameterisation seem to do better than the simpler parameterisations for "Cold0". Also the plume emulator seems to do quite okay for the "Cold1" case.

We already say that the plume emulator does good for the "Cold_0" scenario. For the rest, It is not that obvious to me so I prefer to keep the text as it is.

page 23, line 16: Interesting to add: if you look at Figure 6, e.g., during the melt pulse in the beginning, the order seems to be reversed and melting decreases with the number of boxes.

Good remark, we added this point

page 25, line 2: "Plume parameterisation configuration" instead of "coupled model configuration"?

Corrected

Technical comments:

Page 17, line 17: get

Corrected

Page 21, line 1: compared

Corrected

Page 25, line 33: idealized

We used British English all over the text